# Investigation of Very Large Eddy Simulation Method for Applications of Supersonic Turbulent Combustion

Chong Yan , Yibing Xu, Ruizhe Cao and Ying Piao *

School of Aerospace Engineering, Tsinghua University, Beijing 100084, China
* Correspondence: piaoying@mail.tsinghua.edu.cn

**Abstract:** The very large eddy simulation (VLES) method was investigated for supersonic reacting flows in the present work. The advantages and characteristics of the VLES model and the widely used improved delayed detached eddy simulation (IDDES) method were revealed through a supersonic ramped-cavity cold flow. Compared to the IDDES model, the VLES model transformed from RANS mode to LES mode faster, resulting in a smaller gray region caused by the mode transition. However, the original volume-averaging truncation length scale could lead to poor predictions of the velocity profiles and wall pressure distribution. By introducing a hybrid truncation length scale combining the maximum grid length and the shear layer adaptive (SLA) length with different coefficients, the accuracy of the VLES method was significantly improved, and the issue of the low shear layer position was solved. Moreover, owing to the resolution control function, the VLES method could adaptively model more turbulent kinetic energy and maintain a good accuracy in a coarser mesh. Finally, the modified VLES method was applied in conjunction with a hybrid combustion model constructed by the partially stirred reactor (PaSR) model and the Ingenito supersonic combustion model (ISCM) in simulations of the supersonic flame in the DLR scramjet combustor. After introducing the correction of the molecular collision frequency by the ISCM, the results obtained by the hybrid combustion model were more consistent with the experimental results, especially for the time-averaging temperature profile in the ignition zone.

**Keywords:** supersonic combustion; very large eddy simulation; finite-rate combustion model; hybrid RANS/LES; high-fidelity simulation

## 1. Introduction

The scramjet is widely seen as the most promising propulsion device for high-speed aircraft, given its comparatively simple structure and ease of maintenance. However, the mixing, ignition, and stable combustion of fuel and air within the limited distance of the chamber at such high speeds remain critical issues. To address this, numerous experimental and numerical simulation studies of ramjet combustors with various flame stabilization devices have been conducted worldwide in the past several decades [1–6].

Throughout the entire flow path of a scramjet, unsteady phenomena have significant implications for many flow and flame characteristics, such as intermittent ignition events, vortex dynamics, and shock/boundary/flame interactions. As a result, a high-fidelity numerical simulation is necessary to uncover the intricate physicochemical couplings between chemical kinetics and supersonic flows. However, for industrial applications, computationally resolving all the relevant spatial-temporal scales in high-speed compressible reactive flows, especially for the boundary layers, is unaffordable when employing a wall-resolved large eddy simulation (LES) method [7], not to mention the direct numerical simulation (DNS) method. In this situation, the limitations of Reynolds-averaged Navier–Stokes simulations (RANSs) and the excessive computational expense of a wall-resolved LES make the use of the hybrid turbulence methods a compromise [8]. Some prior work in applying hybrid RANS/LES methods for simulations of high-fidelity supersonic turbulent

combustion has been reported [9–11], wherein the boundary layers along with the vortex from the mixing layers were adequately resolved.

There are various techniques to resolve the blending of turbulence models. According to their baseline model, the hybrid RANS/LES models can be categorized into three groups [12]: RANS-based models, including the detached eddy simulation (DES) [13–15] method, the very large eddy simulation (VLES) [16] method, and the partial averaged Navier–Stokes equation (PANS) method [17,18]; LES-based models, such as wall-modeled LES (WMLES) and Reynolds-stress-constrained large eddy simulation (RSC-LES) [19]; and coefficient average methods, wherein a wall-distance-dependent function acts as the blending function of the RANS and LES turbulent viscosity [20,21]. The latter two categories of the methods require more grids to ensure the quality of the simulation results, since the LES model is applied directly to simulate most of the flow field except the near-wall regions. Considering the computational cost, this paper mainly focuses on the performance of the RANS-based hybrid methods in supersonic flows.

The DES method and its modifications have been successfully applied to a wide range of problems, from validation cases to complex engineering applications [22–25]. To address the main problems of the original DES model, such as modeled stress diminishing (MSD) and grid-induced separation (GIS), the improved delayed detached eddy simulation (IDDES) method [15] adopted a new grid length scale and introduced a delaying function and an elevating function into the mixing length scale. To capture the Kelvin–Helmholtz (K-H) instability phenomena in the shear layers with higher resolution, Shur [26] proposed the shear-layer adapted (SLA) length scale for the IDDES method to replace the upper limit of the grid scale. Despite some minor flaws, such as not being completely formulated for cases involving complex geometry, introducing too many model parameters, and having a gray region, IDDES is still a well-performing and reliable hybrid RANS/LES model for engineering applications.

At nearly the same time, Speziale [16] proposed the very large eddy simulation (VLES) method, wherein the turbulent viscosity of the Reynolds-averaged Navier–Stokes (RANS) model was attenuated according to the analysis of the turbulent energy spectrum and the local length scales. This approach enabled a smooth evolution from RANS to quasi-direct numerical simulation (quasi-DNS) for all grids. However, Speziale's original VLES model attenuated the Reynolds stress too quickly, making it nearly impossible to return to a RANS simulation unless the mesh was unreasonably coarse. To improve the VLES method's performance in the near-wall regions, Han and Krajnović [27] proposed a new form of the resolution control function by introducing the turbulent integral length scale. The improved VLES (IVLES) model was applied to some basic RANS models, such as the standard $k - \varepsilon$ model [28] and $k - \omega$ model [29], for many tests, and generally good results were obtained for some incompressible or subsonic flows [30–33]. The VLES method's adaptive resolution control function allowed it to outperform most of the LES models in relatively coarse meshes [27,30,34,35]. Additionally, the resolution control function in the VLES model is independent of the wall distance, making it more suitable for engineering applications with complex configurations and turbulent structures.

Besides the turbulence model, the discretization scheme is another important factor in the ability to capture small flow features. In numerical simulations involving strong discontinuity such as shockwaves, total variation diminishing (TVD) schemes are always used to avoid numerical oscillations and improve the robustness of the simulation. However, additional numerical viscosity is always induced by the gradient limiters of the TVD-based schemes. In contrast, the high-order central schemes with lower dissipation are better at capturing the flow structures of small vortices but are less unstable around the shockwaves. For this reason, many shock-capturing functions for low-dissipation blending schemes, such as the Harten switch [36], Pirozzoli switch [37], Ducros sensor [38], and Hill–Pullin shock sensor [39] have been proposed for distinguishing the shockwave regions. Among these methods, the Ducros blending function is easy to calculate and performs well in simulations of supersonic flows with shock/boundary interactions [9,10]. However, the

expansion fans can also be judged as the TVD region under the original Ducros sensor, which may bring additional dissipation to the heat-release regions with high expansion rates in combustion cases.

As a RANS-based hybrid turbulence model, the choice of baseline model has a great impact on the performance of the VLES model in specific cases. Xia compared three different RANS models in simulations of strongly swirling turbulent flow and found that the VLES method based on the baseline (BSL) $k - \omega$ RANS model had the best performance for a high-Reynolds-number complex flow [34]. In the present work, the $k - \omega$ shear-stress transport (SST) model [40], along with compressible dissipation corrections [41] and structural compressibility corrections [42], was employed as the baseline RANS model for better performance in supersonic flows.

This paper is organized as follows. The numerical details of the VLES method are presented first, following the descriptions of the combustion model and discretization scheme. Then, the computational results obtained from the present VLES and IDDES models of a supersonic flow over a ramped cavity [43], as well as the comparisons with available experimental data and the RANS predictions, are presented, with a comparative study of different grid length scale estimation methods in the VLES model. The supersonic reacting flow in a hydrogen-fueled strut-injection scramjet model combustor [1] was investigated numerically using the VLES turbulence model with a flame stabilization analysis [44]. Finally, the performance of the proposed VLES method in supersonic turbulent combustion flows is summarized.

## 2. Numerical Methodology

### 2.1. Governing Equations

The governing equations for the multicomponent compressible reactive flow [45], including the conservation laws of the mass, the momentum, the mass of the specie, and the total energy, have the same form in both RANS and LES simulations, which are as follows:

$$\frac{\partial \bar{\rho}}{\partial t} + \frac{\partial \bar{\rho} \tilde{u}_j}{\partial x_j} = 0 \tag{1}$$

$$\frac{\partial \bar{\rho} \tilde{u}_i}{\partial t} + \frac{\partial \bar{\rho} \tilde{u}_i \tilde{u}_j}{\partial x_j} = -\frac{\partial \bar{p}}{\partial x_i} + \frac{\partial}{\partial x_j} \left( \bar{\tau}_{ij} + \tau_{ij}^t \right) \tag{2}$$

$$\frac{\partial \bar{\rho} \tilde{Y}_m}{\partial t} + \frac{\partial \bar{\rho} \tilde{Y}_m \tilde{u}_j}{\partial x_j} = \frac{\partial}{\partial x_j} \left[ \left( \bar{\rho} D_m + \frac{\mu_t}{Sc_t} \right) \frac{\partial \tilde{Y}_m}{\partial x_j} \right] + \tilde{\omega}_m \tag{3}$$

$$\frac{\partial \bar{\rho} \tilde{E}}{\partial t} + \frac{\partial \bar{\rho} \tilde{H} \tilde{u}_j}{\partial x_j} = \frac{\partial}{\partial x_j} \left[ \kappa \frac{\partial \tilde{T}}{\partial x_j} + \frac{\mu_t}{Pr_t} \frac{\partial \tilde{h}}{\partial x_j} \right]$$
$$+ \frac{\partial}{\partial x_j} \left[ \sum_{m=1}^{ns} \left( \bar{\rho} D_m \tilde{h}_m \frac{\partial \tilde{Y}_m}{\partial x_j} \right) + \tilde{u}_i \left( \bar{\tau}_{ij} + \tau_{ij}^t \right) + \left( \mu + \frac{\mu_t}{\sigma_k} \right) \frac{\partial k}{\partial x_j} \right] \tag{4}$$

In the above equations, the variables $\rho, u, p, T$ are density, velocity, pressure, and temperature, respectively. The total energy and enthalpy are $E = e + K$ and $H = e + p/\rho + K = h + K$, respectively, where $e$ is the sensible and chemical internal energy with the mass formation enthalpy so that the heat release term does not appear on the right hand side of Equation (4). The variables $Y_m$ and $h_m$ are the species mass fraction and sensible enthalpy of specie $m$, $\dot{\omega}$ is the reaction rate, $\kappa$ is the Fourier coefficient of thermal conductivity, and $D_m$ is the mass diffusion coefficient of the $m$th specie. For accurate mass diffusion predictions in multicomponent flow, the fourth-order log-polynomial of temperature for the binary mass diffusion coefficient $D_{ij}$ and the algorithm for estimating the individual diffusion

coefficient $D_m$ from the CHEMKIN code [46] were used in this study. The molecular and turbulent viscous tensors are:

$$\bar{\tau}_{ij} + \bar{\tau}_{ij}^t = 2(\mu + \mu^t)\tilde{S}_{ij} - \frac{2}{3}(\mu + \mu^t)\tilde{S}_{kk}\delta_{ij} - \frac{2}{3}\bar{\rho}k\delta_{ij} \tag{5}$$

In the RANS framework, the superscripts ⁻ and ˜ denote the ensemble and Favre averaged quantities, respectively, while in the LES framework, they represent the filtered and Favre filtered quantities. From another point of view, all the turbulent kinetic energy ($k$) is modeled in RANS, while only the subgrid-scale $k$ is modeled in LES [47]. In the VLES methodology, the RANS stress is attenuated to an appropriate value using the resolution control function ($F_r$) to resolve $k$ as much as possible according to the local grid length scale and turbulent length scales.

### 2.2. Shear-Stress Transport Model

According to extensive experiences using the RANS models, the shear-stress transport (SST) model [40] offers several advantages compared to the standard $k - \omega$ model and $k - \varepsilon$ model. Firstly, the SST model performs better than the $k - \omega$ model [29] for high-Reynold-number shear flows by introducing the cross-diffusion term from the $k - \varepsilon$ model. The transport of the shear stress in the boundary layers is also considered in the SST model by adding a limiter to the turbulent viscosity so that the performance of the SST model in the turbulent boundary layers with adverse pressure gradients is improved. In the present work, the hybrid RANS/LES models were all formulated based on Menter's $k - \omega$ shear-stress transport (SST) model, considering the advantages of the SST model in supersonic flows with complex turbulent structures. The expression of turbulent viscosity $mu_t$ and the governing equations of the turbulent kinetic energy $k$ and specific turbulent frequency $\omega$ in the SST model are given as follows:

$$\frac{\partial \bar{\rho}k}{\partial t} + \frac{\partial \bar{\rho}k\tilde{u}_j}{\partial x_j} = \alpha_1 P_k + \frac{\partial}{\partial x_j}\left[\left(\mu + \frac{\mu_t}{\sigma_k}\right)\frac{\partial k}{\partial x_j}\right] - \alpha_2\beta^*\bar{\rho}k\omega \tag{6}$$

$$\frac{\partial \bar{\rho}\omega}{\partial t} + \frac{\partial \bar{\rho}\omega\tilde{u}_j}{\partial x_j} = \frac{\gamma\bar{\rho}P_k}{\mu_t} + \frac{\partial}{\partial x_j}\left[\left(\mu + \frac{\mu_t}{\sigma_\omega}\right)\frac{\partial \omega}{\partial x_j}\right] + \frac{2\bar{\rho}\sigma_{\omega 2}}{\omega}(1 - F_1)\frac{\partial k}{\partial x_j}\frac{\partial \omega}{\partial x_j} - \beta\bar{\rho}\omega^2 \tag{7}$$

$$\mu_t = \frac{a_1\bar{\rho}k}{\max(a_1\omega, \tilde{S}F_2)} \tag{8}$$

where $a_1, \gamma, \beta, \sigma_k,$ and $\sigma_\omega$ are the model parameters of SST; $\tilde{S}$ is the invariant measure of the strain rate; and $F_1$ and $F_2$ are the first and second blending functions that equal unity on wall boundaries and rapidly switch to zero away from boundary layers.

In supersonic flows, it is essential to correct the turbulence model considering the compressible effect, especially for the outer layer with the $k - \varepsilon$ mode in the SST model. Sakar [41] provided the correction factors $\alpha_1$ and $\alpha_2$ of the pressure-dilatation compressibility effect for the $k - \varepsilon$ model. Introducing the turbulent Mach number $M_t = \frac{\sqrt{2k}}{\tilde{a}}$, they are written as:

$$\alpha_1 = 1 - 0.4(1 - F_1)M_t^2 \tag{9}$$

$$\alpha_2 = 1 + 0.8(1 - F_1)M_t^2 \tag{10}$$

### 2.3. Very Large Eddy Simulation

As for the very large eddy simulation (VLES) method, the resolution control function $F_r$ can be applied to various RANS models in the same way. A comparative study of the performance of the VLES method based on the standard $k - \omega$ model, $k - \varepsilon$ model, and baseline $k - \omega$ model has already been carried out in a strongly swirling turbulent flow

using an incompressible solver [34]. The results showed that the VLES BSL $k - \omega$ model performed better than the others in predicting the various fundamental flow mechanisms, such as vortex breakdown, precessing vortex core, and K-H instability. In the present work, the VLES model was formulated based on Menter's $k - \omega$ shear-stress transport (SST) model, using the improved resolution control function from Han [27]. The expression of turbulent viscosity was modified as follows:

$$\mu_t = \frac{F_r a_1 \bar{\rho} k}{\max\left(a_1 \omega, \tilde{S} F_2\right)} \tag{11}$$

Furthermore, the resolution control function $F_r$ was induced to damp the turbulent viscosity away from the walls, which is the core of VLES modeling, expressed as follows:

$$F_r = \min\left(1.0, \left[\frac{1.0 - \exp(-\beta_{VLES} L_c / L_k)}{1.0 - \exp(-\beta_{VLES} L_i / L_k)}\right]^2\right) \tag{12}$$

$$L_c = C_x \Delta, L_i = \frac{k^{3/2}}{\beta^* k \omega}, L_k = \frac{v^{3/4}}{(\beta^* k \omega)^{1/4}} \tag{13}$$

where $\beta_{VLES} = 0.002$ is the model parameter, and $L_c, L_i$, and $L_k$ are the cutoff length scale, turbulent integral length scale, and Kolmogorov length scale, respectively. According to the asymptotic analysis of Equation (12), $C_x = \sqrt{0.3 C_s / \beta^*}$ is a constant related to the Smagorinsky model constant $C_s$. When the standard Smagorinsky model constant $C_{s,0} = 0.1$ is taken, $C_x = 0.61$. A generic and easy-to-use modification of $C_s$ based on a theoretical analysis [48] could also be used:

$$C_s = \sqrt{\frac{\left[\left(C_{s,0}^2 \tilde{S}\right)^2 + v^2\right]^{1/2} - v}{\Delta^2 \tilde{S}}} \tag{14}$$

In the VLES model of Han [27], the subgrid length scale is estimated by the cube root of the mesh volume, $\Delta_{ave} = \left(\Delta_x \Delta_y \Delta_z\right)^{1/3}$, which is reasonable for isotropic meshes in the LES region but risky for anisotropic meshes whose wall-normal grid length is far smaller than the other two directions in the near-wall regions. In supersonic turbulence flows, the height of the grid in the first level is even lower in the wall-normal direction, because of the high speed of the entrance condition. Additionally, the boundary layer thickness is also smaller than that of incompressible subsonic flows. Therefore, the underestimation of the grid size in the supersonic boundary layer using RANS-like meshes is more likely to cause the problem of modeled stress diminishing (MSD). Considering this situation, the idea of delay separation in the IDDES model is quite enlightening. In the resolution control function of VLES, two length scales (the integral length scale and the Kolmogorov length scale) have been set up. Only the calculation method of the truncation scale has not been determined. In the present work, a blending grid length scale was proposed to insure that the RANS region near the wall was large enough.

$$L_c = F_1 C_{x,k-\omega} \Delta_{max} + (1 - F_1) C_{x,k-\varepsilon} \Delta_{\boldsymbol{\omega}} \tag{15}$$

$$\Delta_{\boldsymbol{\omega}} = \frac{1}{\sqrt{3}} \max_{n,m=1,8} \left|\boldsymbol{n_\omega} \times (\boldsymbol{r_n} - \boldsymbol{r_m})\right| \tag{16}$$

where $\Delta_{\boldsymbol{\omega}}$ is the basis of the shear layer adaptive (SLA) length scale considering the unit vector parallel to the curl direction [49,50]; $\boldsymbol{n_\omega}$, $\Delta_{max}$ is the maximum grid scale; $F_1$ is the first blending function of the SST model; and $C_{x,k-\omega} = 0.8, C_{x,k-\varepsilon} = 0.61$ are the coefficients of the cut-off length scale for the inner and outer layers of different turbulence modes. With this calculation method, the value of the grid length scale in the $k - \omega$ mode increases, and quickly switches to $C_{x,k-\varepsilon} \Delta_{\boldsymbol{\omega}}$ in the $k - \varepsilon$ mode region.

For the shear layer close to the wall region, it is difficult to convert the mesh into an isotropic type immediately. In this situation, the grid-scale calculation method considering the vorticity direction can reasonably predict the grid scale of the plane where the vortex is expanded, thus avoiding excessive turbulent viscosity, which may lead to the poor prediction of the shear layer instability effect.

### 2.4. Improved Delayed Detached Eddy Simulation

In the original detached eddy simulation (DES) method, the conversion speed to the LES mode away from the wall is too fast [51], resulting in the phenomena of modeled stress diminishing (MSD), grid-induced separation (GIS), and boundary layer logarithmic-law mismatch (LLM). Although the delayed detached eddy simulation (DDES) method alleviates the MSD problem to a certain extent, by introducing a delay factor, the LLM problem is not completely solved. In the improved delayed detached eddy simulation (IDDES) method [15], an elevating factor is introduced based on the DDES method so that the length scale in the near-wall region is further increased. Furthermore, the mismatch phenomenon caused by the inconsistent slopes of RANS and LES during the transition of the log-law region is avoided. In addition, the length scale of the WMLES model is further mixed into the final length scale in the IDDES method, which also enables the model to switch to the WMLES mode when the wall grid is dense.

The IDDES model also uses a unique calculation method for the grid length scale:

$$\Delta_{IDDES} = \min\left\{\max\left[C_w d_w, C_w h_{max}, h_{wn}\right], h_{max}\right\} \tag{17}$$

where $C_w = 0.15$, and $h_{max}, h_{wn}$ represent the maximum scale of the grid and the length of the grid projection in the direction perpendicular to the wall, respectively. Firstly, this calculation method limits the grid scale to $C_w h_{max} < \Delta < h_{max}$. In the viscous sublayer, $\Delta = C_w h_{max}$, considering that the distance from the grid to the wall and the width of the grid in the direction perpendicular to the wall are far smaller than the width of the grid in the flow direction. In the region slightly away from the wall, the anisotropy of the grid does not change much, but the wall distance $d_w$ and the grid span in the direction perpendicular to the wall $h_{wn}$ are all increased. Thus, it is predictable that $\Delta = \max\left[C_w d_w, h_{wn}\right]$. It can be seen that the IDDES method effectively reduces the grid scale near the wall compared to the DDES method, so it is conducive to capturing the small-scale vortex structure in the turbulent boundary layer when the grids are small enough. The calculation method of the final length scale in the IDDES method is defined as follows:

$$l_{IDDES} = \tilde{f}_d(1 + f_e)l_{RANS} + \left(1 - \tilde{f}_d\right)l_{LES} \tag{18}$$

where the coefficient of the LES length scale is related to the basic RANS model:

$$l_{LES} = C_{DES}\Delta_{IDDES}, C_{DES} = F_1 C_{DES,k-\omega} + (1 - F_1)C_{DES,k-\varepsilon} \tag{19}$$

The calibrated model coefficients are $C_{DES,k-\omega} = 0.8$ and $C_{DES,k-\varepsilon} = 0.61$, respectively, and the specific calculation method of the delay factor $\tilde{f}_d$ and the elevating factor $f_e$ can be found in [15].

In the IDDES model, the transport equations of the turbulent kinetic energy and turbulent frequency are similar to those of the compressible SST model outlined in the above section, but only the dissipation term of the $k$ equation is modified, as follows:

$$\varepsilon_k = \rho \frac{k^{\frac{3}{2}}}{l_{IDDES}} \tag{20}$$

### 2.5. Hybrid Partially Stirred Reactor/Ingenito Supersonic Combustion Model

The partially stirred reactor [52] (PaSR) model, in which the subgrid flow structure is divided into the fine structure and large-scale coherent flow structure, was used to estimate

the filtered reaction source term in this study. Moreover, the compressible effect, which has a large impact on the filtered chemical reaction source term in supersonic reacting flow, was also considered through the correction factor proposed by Ingenito [53]. According to his theoretical analysis, the molecular collision coefficient in the Arrhenius formula should be modified considering the thermodynamic non-equilibrium phenomena in supersonic flows. From the microscopic point of view, the high-speed compression effect can reduce the average free path of molecules and increase their collision frequency. Therefore, the correction factor in the present hybrid PaSR-ISCM combustion model was proposed with the following expression:

$$\tilde{\omega}_m = \gamma^* \omega_m \left( \bar{\rho}, \tilde{Y}_1, \tilde{Y}_2, \cdots, \tilde{Y}_{ns-1}, \tilde{T} \right) \tag{21}$$

$$\gamma^* = \left( 1 + 2M_t^2 \right) \frac{\tau_c}{\tau_c + \tau_k} \tag{22}$$

where $\tau_k = C_{mix} \sqrt{\frac{\nu + \nu_t}{\beta^* k \omega}}$ represents the subgrid mixing timescale, and $\tau_c$ is the chemical reaction timescale, which can be understood as the ratio of the molar concentration to the weighted average of the molar production rates of each reaction [54–56]. The details of the calculation process of $\tau_c$ are shown in Appendix A, and the formula for the turbulent Mach number $M_t$ was provided in Section 2.2 for the compressibility correction of the SST model.

The hydrogen combustion chemistry used in this study was the global reaction mechanism from Ó Conaire et al. [57], including 9 species and 21 reactions. According to a series of experiments and numerical simulations, this detailed mechanism applies to the temperature range of 298 K to 2700 K, the pressure range of 0.05 atm to 87 atm, and the equivalence ratio range of 0.2 to 6.

### 2.6. Discretization Method

The governing equations of compressible turbulent reacting flow were solved using a finite-volume block-structured in-house code [22,45], where the convection fluxes were discretized by the Harten Lax and van Leer contact (HLLC) approximate Riemann solver [58], and the viscid fluxes were discretized by the second-order central Jacobian scheme. As for time advance, each time step was split into a convection–diffusion substep and a chemical reaction integration substep [59]. The convection–diffusion substeps were advanced using the second-order implicit lower-upper symmetric Gauss–Seidel (LUSGS) method [60,61]. In the reaction substeps, the governing nonlinear stiff ordinary differential equations (ODEs) for the compositions including the species mass fraction and temperature were integrated under constant volume conditions using the DVODE [62] solver, which guaranteed that the ODE integration error was insignificant compared to the flow discretization errors. As for the reconstruction method, a modified Ducros blending scheme was used to interpolate the primitive variables from the cell center to the face center. The hybrid interpolation schemes based on the modified shock-capturing sensor were combined by the third-order MUSCL scheme [63] with the Venkatakrishnan limiter [64] and the fourth-order central scheme.

The total variation diminishing (TVD) scheme can provide convincing computational robustness since it reduces the numerical oscillations in addition to the pressure discontinuities via the gradient limiters. However, the numerical dissipations of TVD schemes are always higher than those of central differencing schemes. Due to the additional numerical dissipation, many small-scale turbulent vortices in the compressible flows can be dissipated unphysically, and the temporal-spatial resolution in high-fidelity large eddy simulations is greatly reduced. Therefore, the Ducros sensor [38] was proposed to blend the central differencing schemes with the TVD schemes, i.e.,:

$$f = \frac{(\nabla \cdot \tilde{\boldsymbol{u}})^2}{(\nabla \cdot \tilde{\boldsymbol{u}})^2 + (\nabla \times \tilde{\boldsymbol{u}})^2 + \epsilon^2} \tag{23}$$

where $\tilde{\boldsymbol{u}}$ is the velocity vector, and $\epsilon = \frac{10^{-8} u_\infty}{\max\left(\Delta_x, \Delta_y, \Delta_z\right)}$ is the ratio of the incoming velocity to the local truncation. The magnitude of the velocity divergence was determined by the local material derivative of mass density: $\nabla \cdot \tilde{\boldsymbol{u}} = -\frac{1}{\bar{\rho}}\left(\frac{\partial \bar{\rho}}{\partial t} + \tilde{\boldsymbol{u}} \cdot \nabla \bar{\rho}\right) = -\frac{1}{\bar{\rho}} \frac{D\bar{\rho}}{Dt}$. Therefore, the blending function tended to unity around the regions with intense mass density variations induced by strong compression or expansion waves, and it tended to zero around the regions dominated by large-scale turbulent coherent structures. In non-reacting flows, the large-scale structure of turbulent eddies downstream of the shock wave can be systematically detected by the original Ducros blending function and resolved by the central difference schemes. However, in compressible reactive flows, the flow expansions refer to both the rarefaction waves and the intensive heat release. Note that the velocity divergence has a large magnitude around the onset location of the turbulent flames, and so the original Ducros blending function was not able to detect the shockwaves around the flames. To use the TVD scheme only in the discontinued regions caused by shockwaves and the central differencing scheme in the regions of expansion fans and flame fronts, the Ducros function was modified as follows:

$$f = \frac{\max(-\nabla \cdot \tilde{\boldsymbol{u}}, 0)^2}{(\nabla \cdot \tilde{\boldsymbol{u}})^2 + (\nabla \times \tilde{\boldsymbol{u}})^2 + \epsilon^2} \tag{24}$$

Since the density of the fluid particles decreases significantly when passing through the flame onset location, the velocity divergence is positive around the flame fronts, and the modified blending function is close to zero. By applying the modification, the numerical dissipation in the flame regions could be further reduced.

## 3. Results and Discussion

The hybrid RANS/LES turbulence models were first validated and compared in a non-reacting supersonic flow through a ramped cavity. The advantages and disadvantages of the different hybrid turbulence models (VLES and IDDES) in supersonic flows along with different calculation methods for the subgrid length scale are compared in detail in this section. Then, the supersonic flame in a strut-based scramjet combustor is investigated using the VLES turbulence model and the partially stirred reactor model considering Ingenito's compressibility correction (PaSR-ISCM).

### 3.1. Case 1: Settles Supersonic Ramped Cavity

3.1.1. Case Setup of the Supersonic Ramped Cavity

Settles took the lead in capturing the flow structure generated by the supersonic incoming flow impacting a ramped cavity through experimental means [65]. Figure 1 shows the experimental configuration and the observed approximate flow structure.

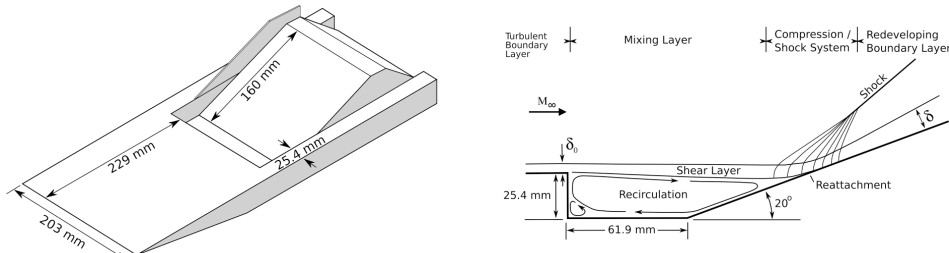

**Figure 1.** Settles configuration and flow field structure diagrams.

The stagnation pressure of the incoming flow is 0.69 MPa, the stagnation temperature is 258 K, and the freestream Mach number is 2.92. In this case, the expansion air with a velocity of about 571.32 m/s passes through a 229 mm long flat plate and a 25.4 mm deep and 61.9 mm long cavity and then impacts on a 20° inclined plane. This case involves many common supersonic flow structures such as a turbulent boundary layer, free shear layer,

reattached boundary layer, and compression shock waves. Therefore, the applicability of the turbulence model in supersonic flow could be comprehensively evaluated based on this case.

A comparative study of the turbulence models was carried out under the same mesh of 4.13 million structural grids. The height of the first layer grid center was $1.5 \times 10^{-6}$ m, which met the requirement that $y^+ < 1$. In the turbulent boundary layer of the flat plate boundary layer in front of the cavity, the grid was only densified in the normal direction of the wall. Therefore, it was not expected that the hybrid RANS/LES method would be able to simulate the turbulent fluctuations in this region to minimize the calculation cost. In the shear layer above the cavity, the grids were densified in both the flow direction and the wall-normal direction, while all grids had the same width in the span direction. Specifically, the number of grid points in the span direction was 33 with a 38.1 mm total width. Thus, the grid length of the spanwise direction was much larger than that of the flow direction and the wall-normal direction for the densified regions such as the boundary layers and shear layers. This posed a great challenge to the estimation of the truncation length scales in the hybrid turbulence model. For the boundary conditions, periodic conditions were adopted for the spanwise interfaces: the exit on the right used the supersonic outlet condition, the upper interface used the far-field non-reflective boundary condition, and the physical wall used the adiabatic non-slip condition. A schematic of the meshes in the spanwise section is shown in Figure 2. To determine the sensitivity of the proposed VLES model with the hybrid grid length scale, the simulations were performed with two different grid resolutions. The fine mesh comprised about 4.1 million structured cells (the flat plate boundary layer was $78 \times 68 \times 32$, the cavity and its upper area were $298 \times 214 \times 32$, and the inclined plane was $281 \times 214 \times 32$), denoted by F or no label, and the relatively coarser mesh consisted of about 1.9 million hexahedral cells (the flat plate boundary layer was $120 \times 80 \times 32$, the cavity and its upper area were $120 \times 201 \times 32$, and the inclined plane was $125 \times 201 \times 32$).

The results of the different turbulence models, including the SST model, the IDDES model with the grid truncation scale $\Delta_{IDDES}$, and the VLES models with two different grid length scales, denoted as VLES-1 (the original volume-averaging grid scale $\Delta_{Vol}$ with a constant coefficient $C_x = 0.61$) and VLES-2 (the novel hybrid form as described in Equation (15)), were firstly compared on the fine mesh.

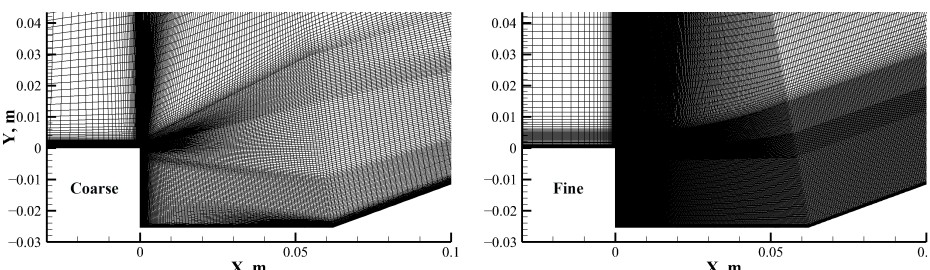

**Figure 2.** Schematic diagram of grid spanwise section of the Settles supersonic flow (**left**: coarse, **right**: fine).

To obtain a suitable initial field, the SST model with compressibility correction was used to obtain the steady RANS result. The LUSGS method was used for the time advance of all equations. The local CFL number was set to 50 for the RANS simulation, while in the hybrid RANS/LES simulations, the global maximum CFL number was set to 15, and the corresponding time step was about $1 \times 10^{-7}$ s, which was similar to that employed by many researches using implicit second-order time-advancing methods [66,67]. For the time averaging of the unsteady hybrid RANS/LES simulations, about 0.1 million time steps were calculated for about eight flow penetration periods.

3.1.2. Validation with Experimental Data of the Supersonic Ramped Cavity

For the validation, the time-averaging streamwise velocity distributions at each section in the numerical simulations and experiments were compared first. All the mean profiles below were obtained by time and spanwise averaging. The first section was located 25.4 mm in front of the cavity, belonging to the fully developed turbulent boundary layer region. Figure 3 shows the comparisons of the mean velocity profiles for this section with the height and velocity normalized in wall units. It can be observed that the velocity distribution in the VLES-1 simulation results showed a too-thick linear distribution region, similar to LES results in over-sparse anisotropy wall grids. This was due to the large aspect ratio of the grids, which could lead to the underestimation of the length scale, resulting in insufficient turbulent viscosity from the LES mode, while the resolving rate of the turbulent kinetic energy was low. Although the differences in the boundary thickness were insignificant, a significant mismatch in the velocity distribution law of the flat-plate boundary layer could lead to poor predictions of the shear layer downstream. In contrast, the IDDES, VLLS-2, and SST turbulence models produced mean velocity distributions in good agreement with the standard boundary layer theory. However, the SST model predicted a slightly lower transition position of the logarithmic law, suggesting that the RANS-predicted turbulent viscosity was too large, resulting in the overprediction of the velocity shear on the wall and a slightly thinner boundary layer.

Figure 4 shows the profiles of the mean velocity normalized by the inflow velocity $U_\infty = 571.32$ m/s at four cross-sections of the different turbulence models, including SST, IDDES, VLES-1, and VLES-2, compared with the experimental data. The thickness of the boundary layer at the first location in the experiment was about 2.9 mm, and the numerical simulation results were in good agreement with the experiment, except for the VLES-1 results. In this section, the attenuation of the turbulent viscosity away from the wall predicted by VLES-1 was too fast, as was also described in the paragraph above. This was mainly caused by the bad estimations of the anisotropic grids' length scales. In the boundary layer region of the flat plate in front of the cavity, the grid was only densified in the direction perpendicular to the wall. This kind of grid is suitable for the RANS, but cannot resolve most of the velocity fluctuations for the LES. Benefiting from the appropriate grid truncation scale, the IDDES method mainly uses the RANS turbulence mode in this kind of wall grid. However, the volume-averaging length scale in the original VLES method is not suitable for the anisotropic grid near the wall, and the premature reduction of the turbulent viscosity easily leads to insufficient modeled stress. For the velocity distributions of the three cross-sections in the shear layer, the results of the IDDES model were in good agreement with the experimental results. Due to the high turbulence viscosity of the SST model, the flow velocity changed too slowly with the increase in height, but the center position of the shear layer was roughly the same as that of the experiment. In VLES-1, the shear layer developed toward a lower direction. This was mainly caused by the deviation in the velocity profile in the flat-plate boundary layer.

According to the VLES-2 results, the attenuation speed of RANS turbulent viscosity became slower away from the wall, and the velocity profile in the flat-plate boundary layer was also consistent with the theory and experiment. By modifying the length scale $L_c$ in the resolution control function, the phenomenon of the too-low shear layer position also almost disappeared. For the second section, the mean velocity predicted by VLES-2 was only slightly higher than that of IDDES in the lower part of the shear layer and tended to be the same as that of IDDES in the upper part of the shear layer. At the third and fourth section positions, the velocity profiles in the shear layer predicted by IDDES and VLES-2 were all in good agreement with the experiment. It can be seen that the deviation in the shear layer development in the original VLES-1 model was effectively solved by introducing the improved hybrid length scale.

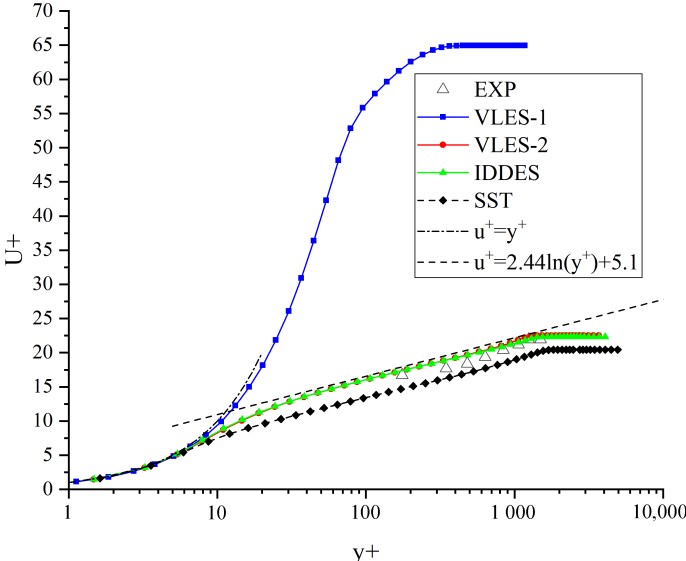

**Figure 3.** Comparisons of mean streamwise velocity profiles for the first section in the wall units.

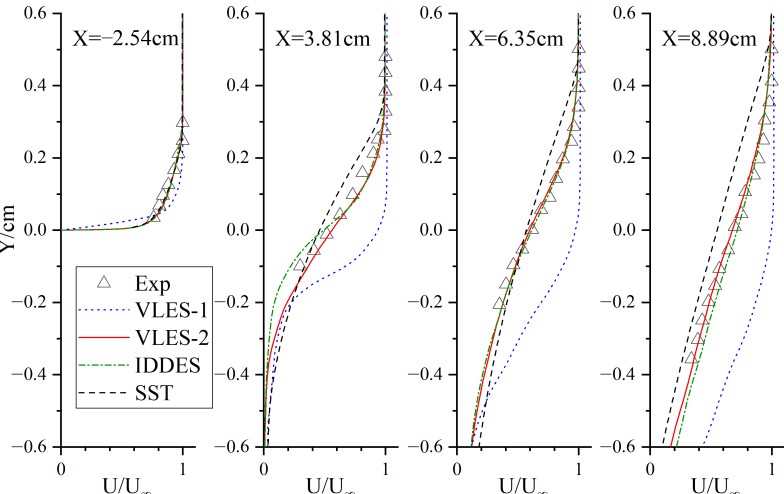

**Figure 4.** Comparisons of mean streamwise velocity profiles at each section.

Figure 5 shows the distributions of the averaging wall pressure normalized by the inflow pressure $P_\infty = 21{,}222.23$ Pa and the experimental data with error bars. The final pressure peak on the slope predicted by the numerical simulation was equivalent to that of the experiment. The RANS simulation using the SST model predicted the most forward pressure rise position, the slowest pressure rise speed, and a later position to reach the peak value. The pressure rise position predicted by the VLES-1 model was in good agreement with the experiment, but the slope of the pressure rise curve was too large, and the peak position was also much earlier than in the experiment. This was mainly caused by the development direction of the shear layer being too low. By comparing the results of VLES-1 and VLES-2, we could see that the problem of the pressure curve rising too fast was effectively solved by using a better grid length scale calculation method. The pressure curve predicted by IDDES had the highest consistency with the experiment, within which the pressure rise position, the position of the peak, and the slope of the pressure curve had a high consistency with the experiment.

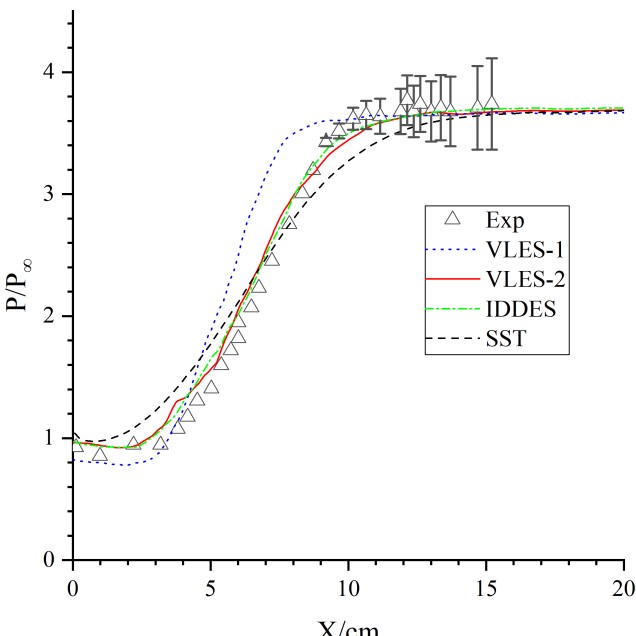

**Figure 5.** Comparisons of wall pressure distributions of different turbulence models.

3.1.3. Analysis of Flow Features

The Q criterion [68] is the second invariant of the velocity gradient tensor, defined as $Q = 0.5[||\Omega||^2 - ||S||^2]$, where $S$ and $\Omega$ are the strain rate tensor and vorticity tensor, respectively. The subgraphs in Figure 6 show the isosurfaces of the Q criterion colored according to the dimensionless streamwise velocity from the simulation results using the IDDES, VLES-1, and VLES-2 turbulence models from top to bottom, respectively. Roughly speaking, the results of IDDES retained the vortex structure of larger scales at the initial stage of the shear layer compared to the VLES results. For the results of VLES with the improved hybrid length scale in particular, the K-H instability phenomenon in the shear layer was effectively captured from a more forward position. Comparing the VLES results using the two different length scale estimations, there was no significant difference between the averaging scales of the vortex in the shear layer. However, it could be seen that in the VLES-2 results, the large deviation in the predicted development direction of the shear layer greatly was improved. The improved VLES results included vortexes with larger scales generated with a higher position in the latter half of the shear layer close to the reattached boundary layer. In addition, the vortex scale in the cavity from the VLES-2 results was slightly larger than in the original VLES-1 results, which indicated that the modified subgrid length scale in the region of near-isotropic turbulence was larger than the volume-averaging grid scale, because the grid scale in the spanwise direction was large. As for the reattached boundary layer region, there was no significant difference between VLES-1 and VLES-2, while the scale of the vortex structure predicted by IDDES was significantly larger than that of VLES.

Figure 7 shows the results of the IDDES turbulence model, including the distributions of the turbulent kinetic energy, the ratio of turbulent viscosity to laminar viscosity, the Mach number, and the static pressure in the center slice. As shown in the distribution of $k$, the turbulence mode used before the cavity was mainly the RANS model. Due to the effect of the transport equations, the viscosity ratio did not decrease rapidly after leaving the wall. In the IDDES results, an obvious gray region was produced just behind the flat plate boundary, resulting in the late occurrence of the instability phenomenon in the initial section of the shear layer. It could be seen from the pressure distribution that the expansion waves and pressure waves reflected in the cavity could be captured using the shock wave capturing reconstruction scheme and the HLLC approximate Riemann solver.

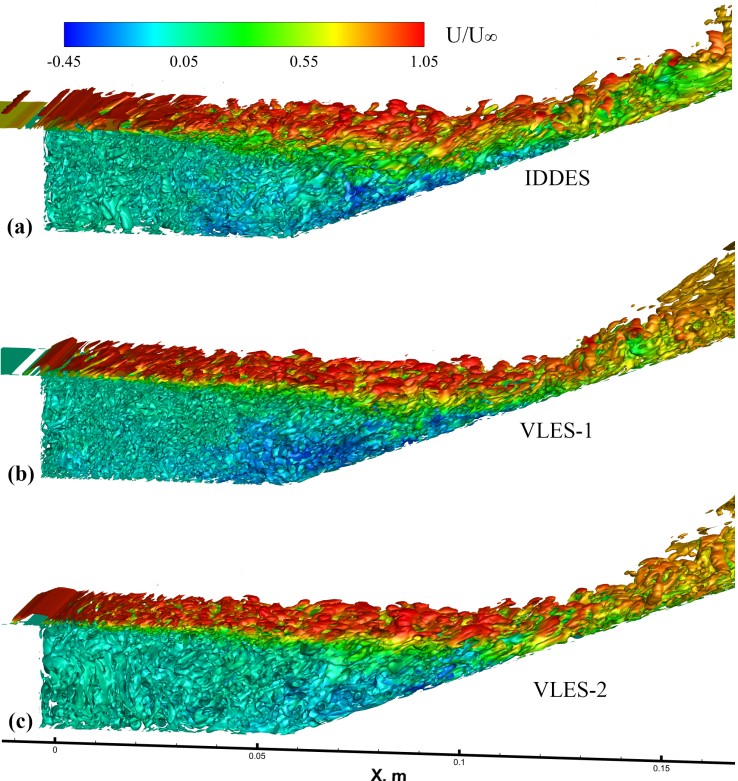

**Figure 6.** Isosurfaces of Q criterion colored according to the dimensionless streamwise velocity from the (**a**) IDDES, (**b**) VLES-1, and (**c**) VLES-2 results.

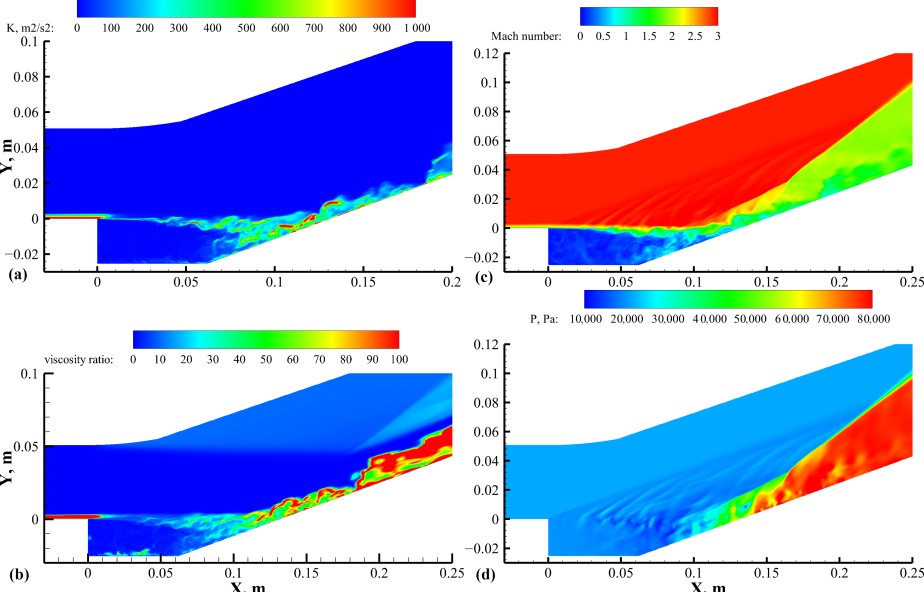

**Figure 7.** Contours of (**a**) turbulent kinetic energy, (**b**) viscosity ratio, (**c**) Mach number, and (**d**) static pressure from the IDDES results.

Figure 8 shows the distributions of the Mach number and viscosity ratio in the center slice of the VLES results using different truncation length scale estimation methods. The subfigures on the left show the results using the volume-averaging grid length scale with the constant cut-off coefficient, while the subfigures on the right show the results using the hybrid truncation length scale, as described in Equation (15). In the Mach number distribution of VLES-1, the shear layer instability phenomenon in the early stage was inconspicuous. This was mainly caused by two factors. Firstly, due to the lack of modeled

stress in the viscous sublayer in the prediction of the velocity profile before the cavity, the velocity gradient at the initial stage of the shear layer was not large enough. Secondly, due to the large width of the grids in the initial shear layer, the length scale estimated by the volume average method was significantly larger than the hybrid length scale considering the correction of the vorticity direction. The essential problem of the VLES-1 results was that the shear layer developed to a lower position, as was also shown in the comparisons of the velocity profiles in Figure 3. Benefiting from the more accurate prediction of the velocity distribution in front of the cavity and smaller length scale in the shear layer, the instability phenomenon of the shear layer was captured more accurately, and the vortex structure behind was also better developed in the VLES-2 results. The predicted velocity profiles in the shear layer region were also in good agreement with the experimental results. It can also be seen from the comparison of the two subfigures at the bottom that the viscosity ratio in the shear layer of VLES-2 was significantly reduced using the hybrid estimation method for the grid length scale. Comparing the viscosity ratio distribution of the IDDES results in Figure 7, it could be found that the viscosity ratio from the RANS mode started to decay significantly at a more forward position in front of the cavity as the grids were densified in the VLES-2 results. The viscosity ratio was directly modified by the resolution control function in the VLES method, while in the DES method, the dissipation term of $k$ was modified to affect the viscosity ratio. Benefiting from this, the gray region in the VLES-2 results was significantly smaller than that in the IDDES results, and the shear layer instability also occurred earlier.

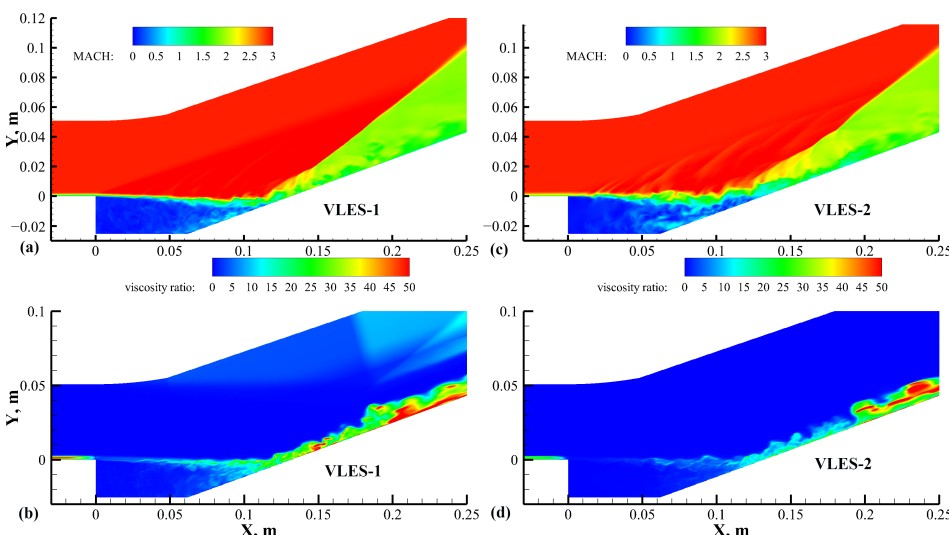

**Figure 8.** Contours of Mach number and viscosity ratio from the VLES results: (**a**,**b**) VLES-1, (**c**,**d**) VLES-2.

The contours of the turbulent kinetic energy $k$ and resolution control function $F_r$ in the VLES results using different length scales are compared in Figure 9. The $k$ value in the VLES-1 results was greater than that in the VLES-2 results, which indicated that the length scale considering the correction of the vorticity direction was more conducive to the capture of small-scale structures in the non-isotropic grid at the initial stage of the shear layer. In the near-wall region, the VLES-2 model used the maximum mesh size with a larger length scale coefficient to ensure that there was sufficient modeled stress for the anisotropic mesh and to avoid the deviation in the velocity distributions caused by the MSD phenomenon.

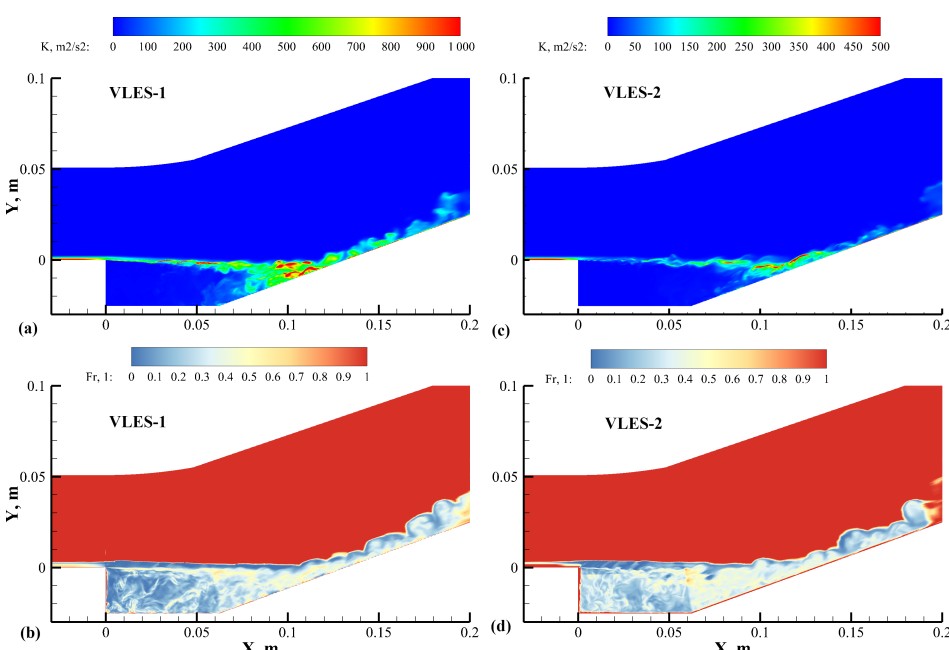

**Figure 9.** Contours of turbulent kinetic energy and resolution control function from the VLES results: (**a**,**b**) VLES-1, (**c**,**d**) VLES-2.

The post-process of the turbulent energy spectrum for the LES was a very effective way to verify that the turbulent kinetic energy was well-resolved. As for the RANS region, where all the turbulent kinetic energy was modeled, and the VLES region, where more than half of the turbulent kinetic energy could not be well-resolved, the turbulent power spectra also showed an insufficiently resolved situation. The turbulent spectra at four positions in the domain are calculated by sampling over time the turbulent kinetic energy with the Discrete Wavelet Transfrom [69] (DWT) method, as shown in Figure 10. This showed that the turbulent kinetic energy at points C and D in the fully developed shear layer and the almost isotropic region inside the cavity were both well-resolved, while the resolved ratio of $k$ at point B in the initial stage of the shear layer was slightly lower, and almost no obvious fluctuations were captured at point A, except for the numerical oscillations. In addition, the slope of the turbulent kinetic energy spectrum for all the statistical points in the inertial subregion followed Kolmogorov's $-5/3$ law, which indicated that the calculation results of the VLES method complied with this basic turbulence statistical law.

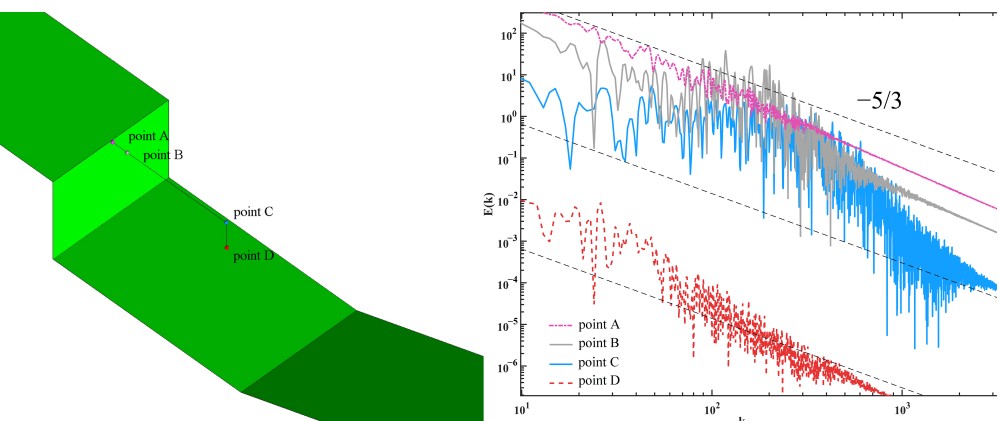

**Figure 10.** Turbulent energy spectrum obtained by sampling over time the $k$ value in various locations of the domain.

### 3.1.4. Grid Sensitivity Test

The very large eddy simulation method was developed for grids that are too coarse to perform a convincing LES or WMLES. The results of various subsonic flows reported by Han et al. [32,34] showed that predictions with good accuracy can be obtained using a low grid resolution with the VLES model. However, whether the good grid sensitivity of the VLES model can be maintained in supersonic flows needs further verification.

Figures 11 and 12 show the time-averaging profiles obtained by the VLES model using the new hybrid grid length scale on two meshes with different grid resolutions, denoted as VLES-2C for the coarse mesh and VLES-2F for the fine mesh. The distributions of the streamwise velocity on several sections and the wall pressure are plotted in Figures 11 and 12, respectively. Remarkably, the present VLES model provided quite similar predictions for the two meshes with significant differences in grid size, especially for the regions of the shear layer and cavity. This implied that good accuracy could be achieved by the current VLES model with a more appropriate length scale, even in a coarse mesh.

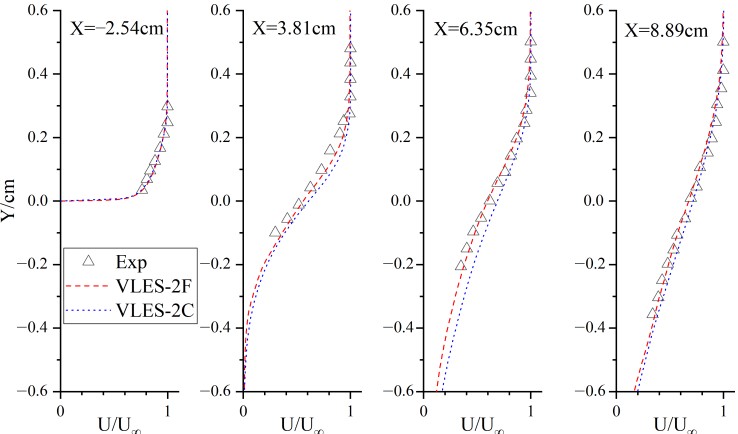

**Figure 11.** Comparisons of mean streamwise velocity profiles on different grids using the VLES-2 model.

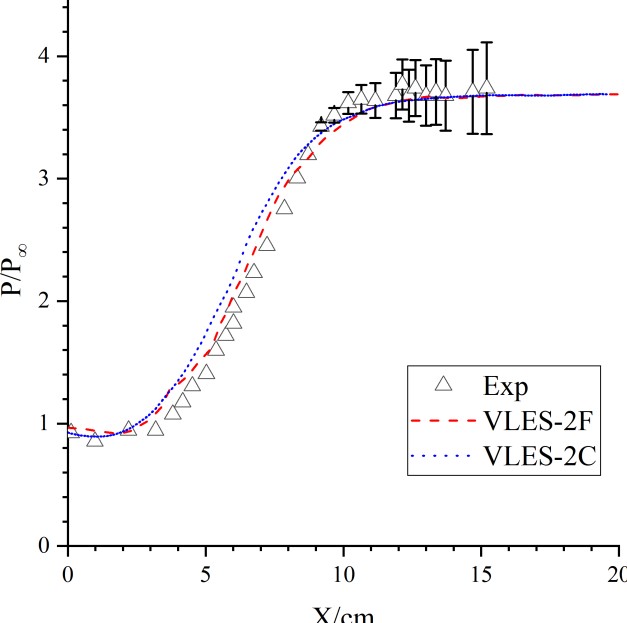

**Figure 12.** Comparisons of wall pressure distributions on different grids using the VLES-2 model.

To further reveal the adaptive adjustment ability of the resolution control function $F_r$ for the resolvable turbulent kinetic energy $k_{res}$, the contours of $F_r$ and the unresolved ratio

of the turbulent kinetic energy $1 - RR_k = \frac{k}{k+k_{res}}$ are shown in Figure 13, where the resolved ratio of $k$ was calculated in the simulations by time averaging, as follows:

$$k_{res} = \frac{\langle \tilde{u}_i \tilde{u}_i \rangle - \langle \tilde{u}_i \rangle \langle \tilde{u}_i \rangle}{2} \tag{25}$$

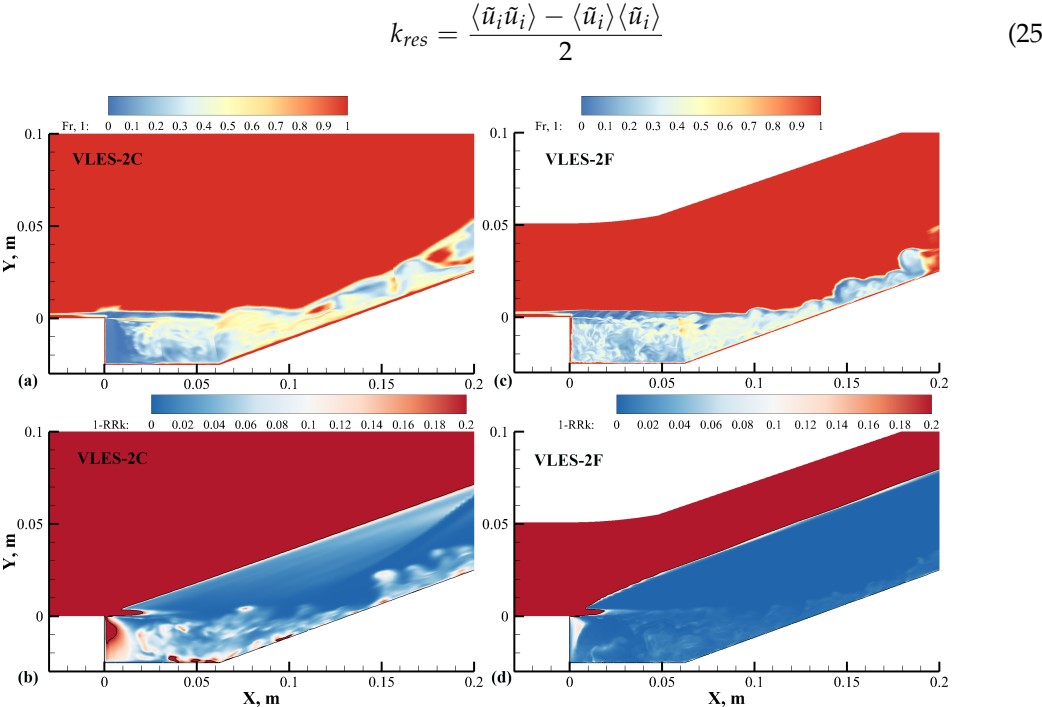

**Figure 13.** Contours of (**a**,**c**) resolution control function and (**b**,**d**) unresolved ratio of turbulent kinetic energy from the VLES-2 results (left: coarse mesh, right: fine mesh).

It can be seen that $F_r$ approached 1 in regions near the wall and far from the main flow structures for both the coarse and fine meshes, indicating that sufficient modeled stress was provided in the wall region where the grids were too coarse in the directions other than the wall-normal direction to resolve most of the velocity fluctuations. The shear layer was the region with the largest turbulent kinetic energy in the simulations, whichever turbulence model was used. In the shear layer, the value of $F_r$ was higher using the coarse mesh than using the fine mesh, which was reasonable, considering the differences in grid scales. In the reattached boundary above the ramped wall, $F_r$ was even higher in many areas using the coarse mesh, and the RANS mode was adopted to maintain accuracy. It should be noted that the unresolved ratio of $k$ is a time-averaging value, while $F_r$ is an instantaneous one. Therefore, the maximum value of the unresolved ratio shown in Figure 13b,d was 0.2 rather than 1. It can be seen from Figure 13b,d that more than 15% of $k$ could not be well-resolved in some places above the cavity and ramped wall under the coarse mesh, while on the fine mesh, more than 90% of $k$ could be resolved in most of the main flow structures, indicating that the VLES model could adaptively change its turbulent viscosity not only for the wall regions but also for the main flow regions to seek a balance between accuracy and fidelity in the simulation results. However, the gray region above the shear layer inherited from the RANS-like flat plate still existed because of the insufficient velocity fluctuations.

### 3.2. Case 2: DLR Supersonic Strut-Based Flame

#### 3.2.1. Case Setup of the Supersonic Strut-Based Flame

The DLR scramjet combustor [1] is shown in Figure 14. In this combustor, vitiated air at a temperature of 340 K flows over a strut. Hydrogen under sonic conditions is injected through fifteen holes uniformly spaced at the base of the wedge-shaped injector. The inlet conditions are provided in Table 1.

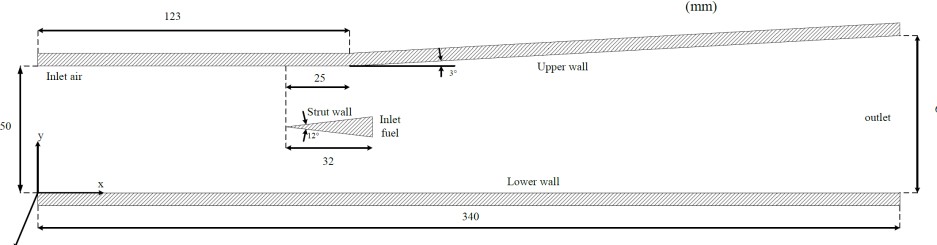

**Figure 14.** Computational set up of the DLR supersonic combustion case.

**Table 1.** Flow conditions of the airstream and the fuel inlet.

|  | Air | Fuel |
|---|---|---|
| U (m/s) | 730 | 1200 |
| T (K) | 340 | 250 |
| P (Pa) | 101,325 | 101,325 |
| $Y_{O_2}$ | 0.232 | 0 |
| $Y_{N_2}$ | 0.736 | 0 |
| $Y_{H_2}$ | 0 | 1 |
| $Y_{H_2O}$ | 0.032 | 0 |

In the present work, the computational domain was 2.4 mm in width, including one fuel injector. Using the RANS model for the near-wall regions, a 3D hexahedron mesh of 2.57 million cells was used for the very large eddy simulations, with the modified truncation length scale following Equation (15). The meshes in the vicinity of the inlet and the near-wall regions were refined. For the boundary conditions, all the physical walls were treated as non-slip and adiabatic. The outflow was assumed to be supersonic so that the values of these ghost cells were extrapolated from the interior of the domain. Periodic conditions were adopted for the interfaces in the spanwise direction.

To study the influence of the different turbulent combustion models and shock wave capturing schemes on the numerical simulation results, three cases are compared in the following section, with their setups summarized in Table 2.

**Table 2.** Case setup for the very large eddy simulations.

| Case Name | Turbulence Model | Combustion Model | Reconstruction |
|---|---|---|---|
| M0 | VLES-2 | PaSR | Modified Ducros |
| M1 | VLES-2 | PaSR-ISCM | Original Ducros |
| M2 | VLES-2 | PaSR-ISCM | Modified Ducros |

### 3.2.2. Validation with Experimental Data of the Supersonic Strut-Based Flame

The simulation results using the modified Ducros scheme (M0, M2) were validated against the measured temperature, streamwise velocity, and its root mean square (rms) fluctuations from experiments. For the comparison with the experimental data, the time-averaging values and their fluctuations were both calculated during the simulations and averaged in the spanwise direction.

Taking the origin of the X-coordinate at the strut base, the time-averaging temperature profiles of the simulation were compared with the experimental data at $X = 11$ mm, 58 mm, and 166 mm, as shown in Figure 15. The profiles of the simulation results from both the PaSR-ISCM and PaSR combustion models are presented. At the first station, two temperature peaks were shown in the simulations and the experiment. These were caused by the heating effect of the shear layer and the heat released from the ignition. For the peak temperatures, both combustion models underestimated the peak value. However, the peak values predicted by the PaSR-ISCM model were higher than the PaSR predictions, which meant that the ISCM could significantly enhance the reaction rate in the shear layer

with a relatively high turbulent Mach number. At the second station, all VLES results showed good agreement with the experimental data, while the PaSR-ISCM predicted a marginally higher temperature peak value in the simulation. The difference between these two combustion models was very small. At the third station, it could be found that the VLES prediction with the PaSR-ISCM combustion model showed better agreement with the experimental data, whereas VLES with PaSR underestimated the peak value of the mean temperature profile. Furthermore, both combustion models overestimated the width of the profile for the last station.

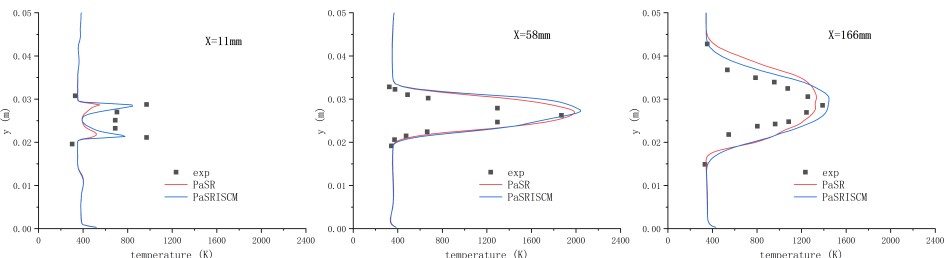

**Figure 15.** Averaged temperature profiles compared with experimental results at three locations using different combustion models.

The experiment also reported the mean streamwise velocity profiles and their fluctuations at several positions downstream of the reacting strut jet. To verify the predicted streamwise velocity distributions, profiles of the mean streamwise velocity and its rms fluctuations from VLES simulations are compared with the experimental results in this section. Figure 16 shows the mean streamwise velocity profiles at three different cross-sections, $X = 11$ mm, 58 mm, and 140 mm. At the first station, both combustion models predicted strong reverse velocity values on each side of the jet behind the strut, and the two predicted reverse peaks of the mean streamwise velocity were much lower than the measured values. Moreover, at the second station, $X = 58$ mm, the predicted mean streamwise velocity profile was narrower than in the experiment, and the minimum mean streamwise velocity value at the center of the profile was also higher than the actual value. However, the two combustion models did not show much difference in the mean streamwise velocity profiles of the first two stations. It can be seen from this figure that good agreement was obtained by the simulations, although the minimum value of the mean streamwise velocity using both combustion models was higher than the measured results. Additionally, the velocity profile of the PaSR-ISCM was flatter than the PaSR result due to the more sufficient heat release predicted with the ISCM correction.

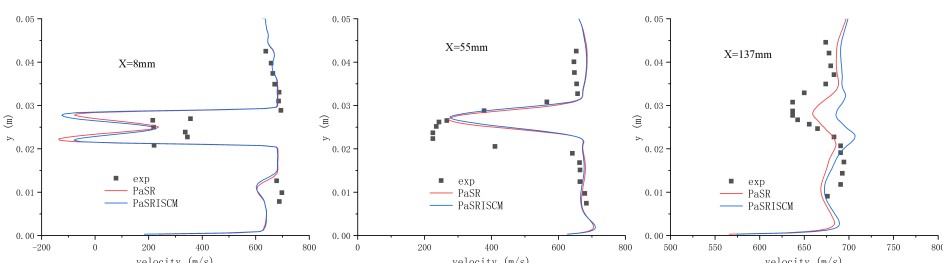

**Figure 16.** Averaged streamwise velocity profiles compared with experimental results at three locations using different combustion models.

Figure 17 presents the profiles of the streamwise velocity fluctuations from the VLES cases at three cross-sections, $X = 11$ mm, 58 mm, and 90 mm. The PaSR-ISCM and PaSR model predictions using the VLES turbulence model showed similar profiles for the streamwise velocity fluctuations of the three stations. At the first station, both simulations overpredicted the velocity rms around the jet. At the second station, the peak value was reduced significantly, and hence reasonable agreement with the experimental data was

obtained for both combustion models. At the last station, both VLES predictions showed almost identical velocity rms profiles. Although the predicted values of velocity rms were slightly higher than was measured for all three stations, the agreements between the VLES predictions and experimental data were reasonable using both combustion models, except for the too-strong reverse flow behind the strut in the simulations.

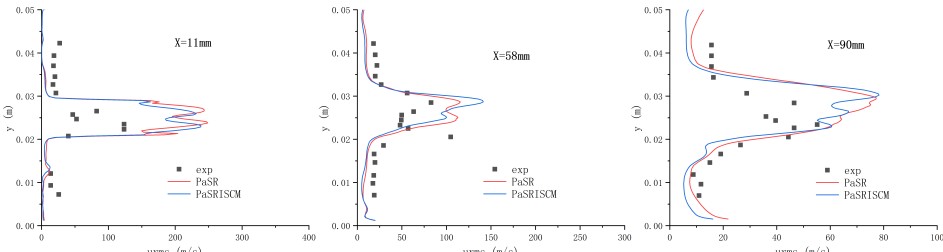

**Figure 17.** Velocity rms fluctuation profiles compared with experimental results at three locations using different combustion models.

The validation with experimental data indicated that the models and schemes adopted, including the VLES turbulence model, PaSR-ISCM combustion model, and the low-dissipation shock-capturing blending schemes, were adequate for predictions of the averaged combustion and flow features in each section of the combustor. It should be noted that the simulation results of the PaSR-ISCM (M2) and PaSR (M0) models, although in good agreement with each other for the streamwise velocity and its rms fluctuations, showed a considerable difference for the mean temperature profile in the first station. Their comparison indicated that the introduction of the ISCM could significantly increase the reaction rate in the high-turbulent-Mach-number regions, which made the result more consistent with the experiment.

### 3.2.3. Analysis of Flame Features

This section discusses the flame features from different points of view in relation to the simulation results. Firstly, the turbulent structures in the mixing and reaction zone behind the strut are illustrated through the isosurface plots of the swirl strength value colored according to temperature in Figure 18.

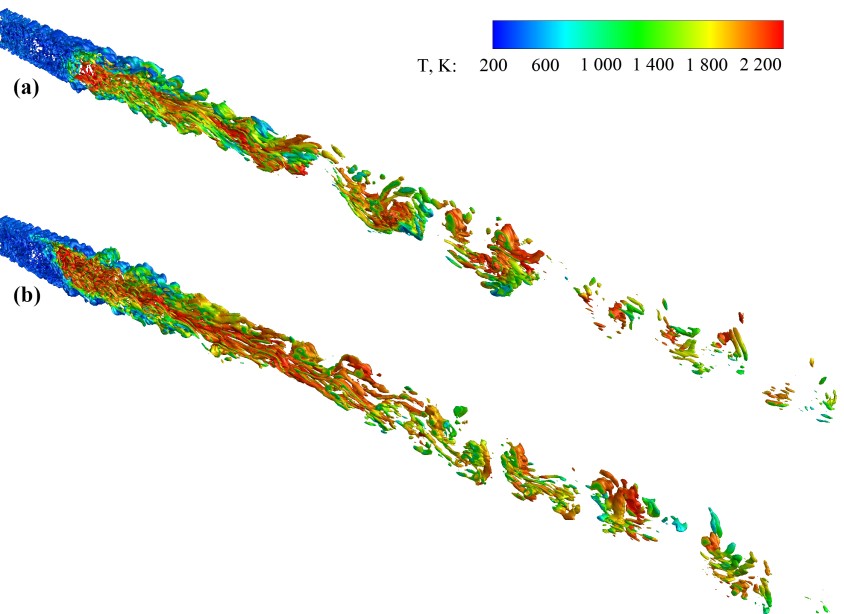

**Figure 18.** Isosurfaces of swirl strength: (**a**) M1, original scheme; (**b**) M2, modified scheme (colored according to temperature).

The swirl strength is defined as the imaginary portion of the complex eigenvalue of the local velocity gradient tensor. It represents the frequency of fluid particles rotating around the vortex core. Higher values of the swirl strength typically correspond to smaller scales of the turbulent structure. Figure 18 shows the isosurfaces of the VLES results for (a) the original scheme (M1) and (b) the modified scheme (M2) using the PaSR-ISCM at the same time, colored according to temperature. The turbulent vortex structure was mainly produced in the mixing zone and the reaction zone behind the strut and near the shearing layer wake. It was also clear that more small-scale turbulent structures were captured with the modified Ducros sensor in the mixing and reaction regions behind the fuel injectors. While the mixture flowed downstream, they developed into vortexes with larger scales of different shapes, and again more small-scale motions were shown in the results of the modified scheme. The numerical dissipation of the simulation in the mixing zone was reduced significantly with the new form of the shock-capturing function. The benefits of the decrease in numerical dissipation could also be seen in the contour plots of other physical fields.

In Figure 19, the simulation results of the original and modified low-dissipation schemes are compared from another perspective. Subplots (a) and (b) show, respectively, the distributions of the blending function in the VLES results using the original (M1) and modified (M2) schemes. In (a), the results of the original scheme with the Ducros blending function were close to unity around the shock waves and the expansion fans on both sides of the combustion region. In contrast, the modified blending function tended to unity only for shock waves on both sides, as shown in Figure 19b. Since the original blending function was nonzero in the expansion and compression regions where the chemical reaction was violent, some additional numerical viscosity was introduced in the mixing zone behind the strut and part of the reaction zone downstream. In contrast, the modified blending function always kept its minimum value in the main flow regions, so that the high-order central scheme was used in the flame expansion region, and the numerical dissipation could be reduced theoretically. The impact of the scheme modification on the temperature distribution is further illustrated in Figure 19c,d. Firstly, the predicted flame onset location using the modified scheme was closer to the strut, which may have been caused by the stronger mixing effect brought about by the higher number of small vortices. As the flame propagated and developed downstream, larger flame kernels were captured by the modified scheme, accompanied by more intense combustion features. Moreover, the internal structure of the flame predicted by the modified scheme was also finer and clearer than the original.

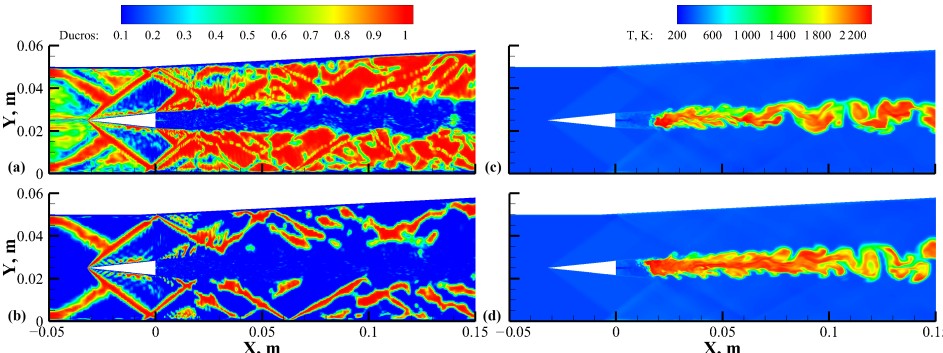

**Figure 19.** Contours of the blending functions of the original and modified low-dissipation schemes (**a**) M1 and (**b**) M2, and the temperature distributions of these two schemes (**c**) M1 and (**d**) M2.

To show the flame structure from a more comprehensive perspective, the mass fraction distributions of the $H$ and $HO_2$ radicals of the original and modified schemes are shown in Figure 20. The production and diffusion processes of the $H$ radical can be seen clearly in Figure 20a,b. In the mixing region of the $H$ radical and the airflow on both sides, clearer vortex structures were simulated by the modified scheme, which made the mixing more

sufficient. It is also clear that the flame wrinkles in the contour plot of the *H* fraction in (b) were more pronounced, which indicated a higher resolution. The production rate of $HO_2$ radicals is higher in low-temperature combustion zones, so its distribution could better reflect the shape of the flame fronts. As shown in Figure 20c,d, the two schemes presented a significant difference in predicting the shape of the flame fronts in the ignition region. Using the modified scheme, the flame front wrinkles were more intense, especially for the strong shearing layers. The new scheme was more accurate for shock wave recognition, so that a high-order scheme with less dissipation could be used in more regions, which significantly improved the fidelity of the simulation.

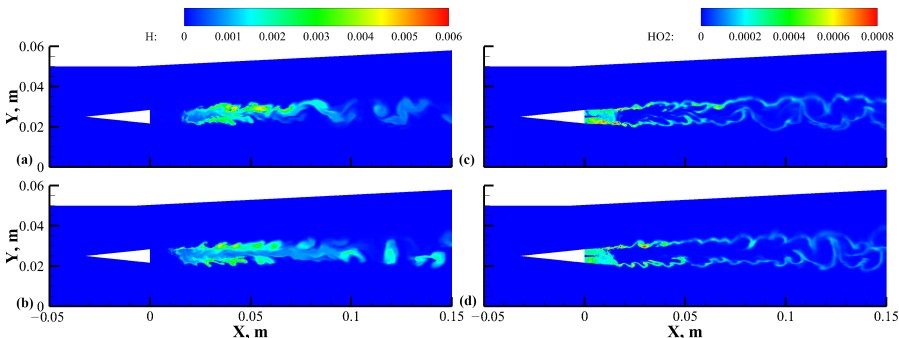

**Figure 20.** Contours of the mass fraction distributions of the *H* radical using the original and modified schemes (**a**) M1 and (**b**) M2; and the mass fraction distributions of the $HO_2$ radical using these two schemes (**c**) M1 and (**d**) M2.

Next, concerning the influences of the combustion model on the reaction rates, Figure 21 compares the results of the original PaSR and the hybrid PaSR-ISCM combustion models using the VLES turbulence model and the modified shock-capturing scheme. Figure 21a,b show the distributions of the filtered reaction rate factor of PaSR and PaSR-ISCM, respectively. As shown in (a), a reduction factor was applied to the reaction rates for which the turbulent timescale was significant under the PaSR model. Comparatively, the multiplier of the reaction rate in the PaSR-ISCM was greater than in regions of a relatively high turbulent Mach number and less than in some vortex cores with a low fine-structure fraction, as shown in (b). Although the PaSR model generally showed good agreement with the experiment in predicting the temperature profiles, the peak temperature of the first station was much lower than measured, which indicated an underestimation of the reaction rate factor. It can be seen from the temperature distribution in (c) that ignition occurred in the shear-layer wake, in which air and fuel were mixed. Farther downstream of the strut, a flame kernel of a high temperature was formulated in the center of the vortices with a relatively low velocity. Taking the partially stirred effect of the turbulence into account, the combustion intensity was lowered significantly. Compared to the original PaSR model, the introduction of the ISCM resulted in a flame onset location closer to the strut and a more intense flame downstream, as shown in Figure 21d. As a consequence of the enhancement of the filtered reaction rate by the ISCM, the flame core with a high temperature formed nearby was also greater in the developed combustion region. Considering the location of the ignition point and the temperature profile of the first section, as shown in Figure 15, the introduction of the ISCM model could increase the accuracy of predicting the filtered chemical reaction rate by considering the influence of the turbulent Mach number on the molecular collision frequency.

Figure 22 shows the net production rates and mass fractions of water using the PaSR and PaSR-ISCM combustion models. In the contours of the water rates, the isoline of the zero value is also plotted to illustrate the dissociation of water. It can be seen that in some high-temperature (over 2200 K) flame cores, the dissociation rates of water were even greater than the production rates, resulting in negative net production rates. Benefiting from the complex timescale estimation method of the chemical reaction system, both the chemical time scale $\tau_c$ and the correction factor $\kappa$ of the PaSR model were small in

these regions. Therefore, the filtered decomposition reaction rate of water was limited to some extent.

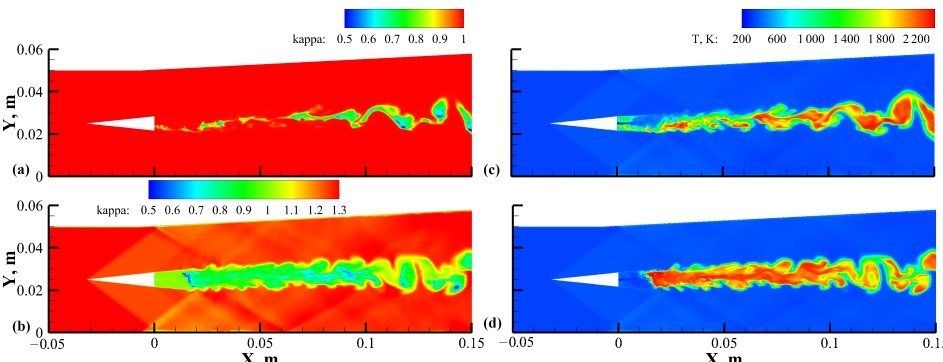

**Figure 21.** Contours of correction factors for the filtered reaction rate using the PaSR and PaSR-ISCM combustion models for (**a**) M0 and (**b**) M2; and temperature distributions using these two combustion models for (**c**) M0 and (**d**) M2.

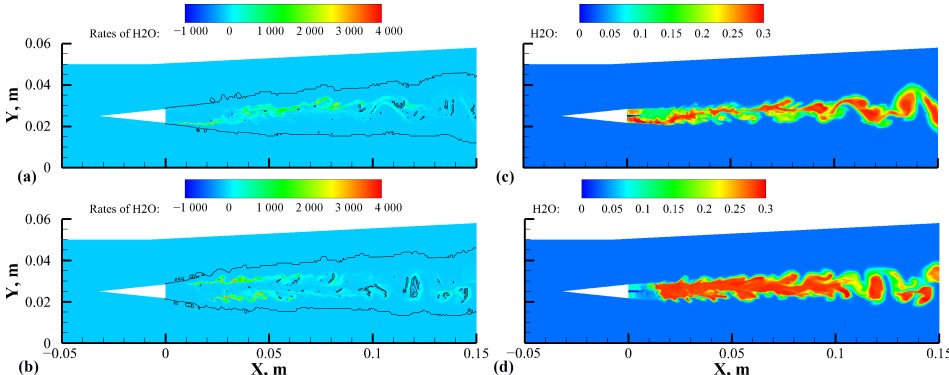

**Figure 22.** Contours of net production rates of water (isoline representing the 0 value) using the PaSR and PaSR-ISCM combustion models for (**a**) M0 and (**b**) M2; and mass fractions of water using these two combustion models for (**c**) M0 and (**d**) M2.

Figure 23 presents the distributions of the time-averaging temperature and its rms fluctuation in the results of the PaSR and PaSR-ISCM models. As shown in Figure 23a,b, the mixture was ignited in the shearing layer behind the strut, and ISCM-PaSR predicted an ignition point closer to the fuel injector and a shorter flame development region. The rms fluctuations from these two combustion models are shown in Figure 23c,d. In the results predicted by the PaSR model, the temperature fluctuation region near the ignition point was also wider. The combustion progress was effectively accelerated by the induction of the ISCM model.

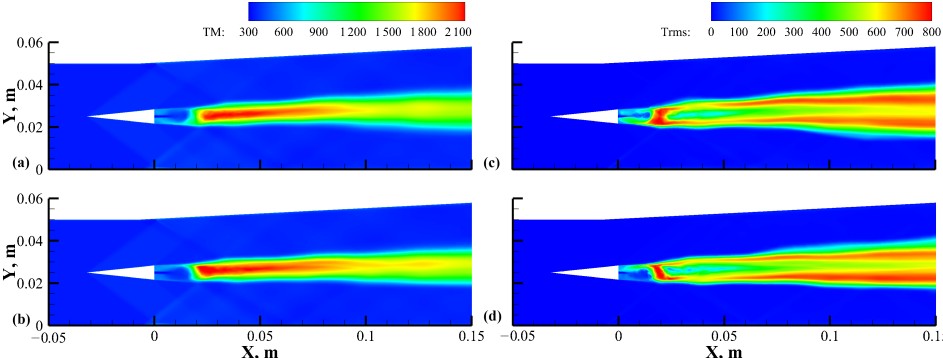

**Figure 23.** Contours of time-averaging temperature using the PaSR and PaSR-ISCM combustion models for (**a**) M0 and (**b**) M2; and rms fluctuation of temperature from the results of (**c**) M0 and (**d**) M2.

### 3.2.4. Performance of the VLES Model

The VLES model was used in the simulations to obtain an acceptable estimation of the boundary layer using a relatively coarse mesh near the wall region and achieve an adaptive turbulence model in the flow field by damping the RANS viscosity to an appropriate value, according to the local flow features. In contrast to other zonal hybrid RANS/LES methods and benefiting from the resolution control method, the modeled turbulent kinetic energy could be included in the VLES framework. Thus, the resolved ratio of $k$, which is very important for the evaluation of turbulent models, could be calculated.

Figure 24a,b show snapshots of the instantaneous resolution control function and the unresolved ratio of $k$ in the simulation of case M2 using the PaSR-ISCM combustion model. As shown in Figure 24a, the resolution control function approached unity near the walls as expected to ensure that the model behaved as a RANS model in the wall regions, while it was approximately zero in the air flow regions on both sides, which indicated that the mesh scale was small enough to directly resolve most of the turbulent structures. For the mixing and combustion regions with complex flow structures, the resolution control function was in between the two limits, providing a VLES (denoted as LES where the preponderance of $k$ was unresolved) or LES adaptive value of the local turbulent viscosity. It can be seen from Figure 24b that the resolved $k$ was larger than 90% in the main flow region downstream of the hydrogen injector, which satisfied the common LES criterion of 80%. The modeled and resolved $k$ values are also shown in Figure 24c,d, respectively. It can be concluded that an adaptive hybrid RANS/LES simulation of supersonic flames was realized by the proposed VLES model, which not only ensured the resolution of the turbulent structure in the mainstream area but also reduced the amount of calculation.

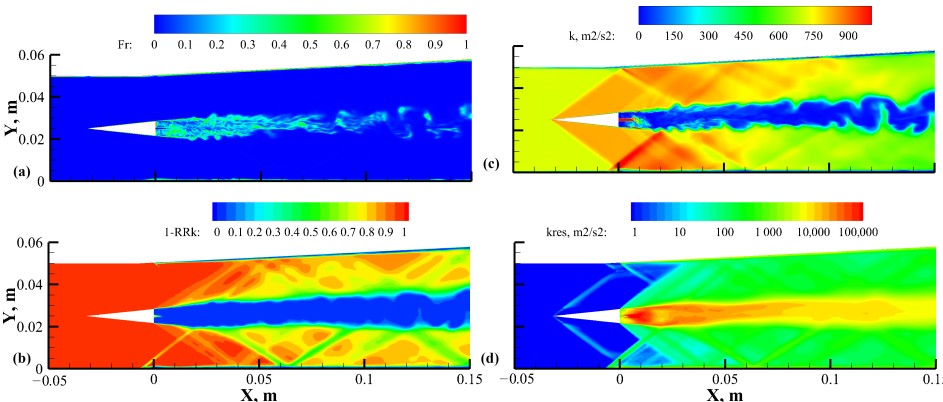

**Figure 24.** Contours of (**a**) resolution control function, (**b**) unresolved ratio of turbulent kinetic energy, (**c**) modeled turbulent kinetic energy, and (**d**) resolved turbulent kinetic energy from case M2 at the spanwise central section.

### 4. Conclusions

In this study, the very large eddy simulation (VLES) method was investigated for supersonic turbulent reacting flows. Firstly, the VLES model was compared to the widely used IDDES model in a Settles supersonic ramped cavity, revealing their advantages and characteristics. In addition, a hybrid truncation length scale was proposed for the VLES model, and comparisons with the original volume-averaging length scale were drawn. Then, the grid sensitivity of the VLES model in a supersonic flow was also tested. As for the reacting flows, a hybrid combustion model combining the partially stirred reactor (PaSR) model and the Ingenito supersonic combustion model (ISCM) was proposed for a better approximation of the filtered reaction rates in the supersonic turbulent flows. To reduce the numerical dissipations and obtain better fidelity, a modification of the Ducros low-dissipation shockwave-capturing scheme for supersonic reacting flows was also proposed and tested. The differences between the proposed hybrid combustion model

and reconstruction scheme and their original forms were shown through the very large eddy simulations of the DLR supersonic flame, and the conclusions are outlined below.

In the VLES results using the original volume-averaging truncation scale, the phenomenon of modeled stress diminishing (MSD) appeared in the flat-plate boundary layer before the cavity, and the shear layer above the cavity developed toward a lower position compared to the results of the other simulations and the experiment. The deviations in the velocity and wall pressure profiles were also very large. A hybrid truncation length scale combining the maximum grid length scale and the shear layer adaptive (SLA) length scale in its simplified form using a blending function with different coefficients were proposed for the VLES model. The accuracy of the velocity and pressure distributions using the hybrid length scale was significantly improved, and the problem of the low shear layer position was also solved. Compared with the IDDES model, transitions from the RANS mode in the boundary layer to the LES mode in the main stream were faster in the VLES model, and the gray region was also smaller. This was mainly because the VLES model directly modified the RANS turbulent viscosity rather than the dissipation term of the turbulent kinetic energy transport equation in the DES models. Benefiting from the resolution control function, the VLES model was not very sensitive to changes in grid density, as shown in the grid sensitivity test. The novel hybrid length scale could delay the mode transition process to improve the accuracy of the VLES model on a coarse wall mesh and reduce the grid length scales in the shear layer by considering the vorticity direction to improve the simulation fidelity.

For very large eddy simulations of the DLR supersonic flame, the introduction of the ISCM method could effectively increase the predicted filtered reaction rate in the regions of a high turbulent Mach number by considering the contribution of the compression effect. Compared with the original PaSR model, the results obtained by the hybrid PaSR-ISCM combustion model were more consistent with the experimental temperature profiles, especially for the first station near the fuel injector. A modification of the original Ducros blending function considering the expansion effect could also reduce the numerical dissipation further in the expansion region of the reactions and the Prandtl–Meyer fans.

Using the VLES turbulence model based on Menter's SST model, in conjunction with the hybrid truncation length scale, a RANS-like estimation of the boundary layers was obtained in a relatively coarse wall mesh. The unresolved ratio of the turbulent kinetic energy also showed that the mixing and combustion regions were mainly simulated by the LES mode. It could be concluded that a high-fidelity simulation of a supersonic flame in the DLR combustor was well-realized with good accuracy and a lower computational cost using the VLES method.

**Author Contributions:** Conceptualization, C.Y. and Y.P.; methodology, C.Y.; software, C.Y. and Y.X.; validation, R.C.; formal analysis, C.Y. and Y.X.; investigation, R.C.; resources, Y.P.; data curation, C.Y. and Y.X.; writing—original draft preparation, C.Y. and Y.P.; writing—review and editing, C.Y., Y.X. and R.C.; visualization, C.Y. and Y.X.; supervision, Y.P.; project administration, Y.P.; funding acquisition, Y.P. All authors have read and agreed to the published version of the manuscript.

**Funding:** This research received no external funding.

**Data Availability Statement:** Not applicable.

**Conflicts of Interest:** The authors declare no conflict of interest.

## Abbreviations and Nomenclatures

The following abbreviations and Nomenclatures are used in this manuscript:

| | |
|---|---|
| VLES | Very large eddy simulation |
| IDDES | Improved delayed detached eddy simulation |
| SLA | Shear layer adaptive |
| PaSR | Partially stirred reactor |
| ISCM | Ingenito supersonic combustion model |
| LES | Large eddy simulation |
| RANS | Reynolds-averaged Navier–Stokes simulation |
| DNS | Direct numerical simulation |
| DES | Detached eddy simulation |
| PANS | Partial averaged Navier–Stokes |
| WMLES | Wall-modeled LES |
| RSC-LES | Reynolds-stress-constrained large eddy simulation |
| MSD | Modeled stress diminishing |
| GIS | Grid-induced separation |
| TVD | Total variation diminishing |
| SST | Shear-stress transport |
| LLM | Logarithmic-law mismatch |
| HLLC | Harten, Lax, and van Leer Contact |
| LUSGS | Lower-upper symmetric Gauss-Seidel |
| ODEs | Ordinary differential equations |
| CFL | Courant–Friedrichs–Lewy |
| DLR | The German Aerospace Center |
| $\rho$ | Mass density, $\mathrm{kg/m^3}$ |
| $u$ | Velocity, m/s |
| $p$ | Pressure, $\mathrm{kg/(m \cdot s^2)}$ |
| $T$ | Temperature, K |
| $E$ | Total energy, $\mathrm{m^2/s^2}$ |
| $e$ | Internal energy, $\mathrm{m^2/s^2}$ |
| $H$ | Total enthalpy, $\mathrm{m^2/s^2}$ |
| $h$ | Internal enthalpy, $\mathrm{m^2/s^2}$ |
| $K$ | Kinetic energy, $\mathrm{m^2/s^2}$ |
| $\tau_{ij}$ | Viscous tensor, $\mathrm{kg/(m \cdot s^2)}$ |
| $\dot{\omega}$ | Mass production rate, $\mathrm{kg/(m^3 \cdot s)}$ |
| $Y$ | Mass fraction |
| $D$ | Mass diffusion coefficient, $\mathrm{m^2/s}$ |
| $\mu$ | Dynamic viscosity, $\mathrm{kg/(m \cdot s)}$ |
| $\nu$ | Kinematic viscosity, $\mathrm{m^2/s}$ |
| $k$ | Turbulent kinetic energy, $\mathrm{m^2/s^2}$ |
| $\omega$ | Specific turbulent frequency, 1/s |
| $\varepsilon$ | Turbulence dissipation rate, $\mathrm{m^2/s^3}$ |
| $Pr$ | Prandtl number |
| $Sc$ | Schmidt number |
| $\kappa$ | Thermal conductivity, $\mathrm{kg \cdot m/(m^3 \cdot K)}$ |
| $\delta$ | Kronecker operator |
| $S_{ij}$ | Strain rate tensor, 1/s |
| $a$ | Sound speed, m/s |
| $M_t$ | Turbulent Mach number |
| $F_r$ | Resolution control function |
| $\Delta_{ave}$ | Volume-averaging length scale, m |
| $\Delta_{max}$ | Maximum length scale, m |
| $\Delta_{\omega}$ | Length scale in the vorticity direction, m |
| $\Delta_{IDDES}$ | IDDES grid length scale, m |

| $d_w$ | Wall distance, m |
| $f_d$ | Delaying factor |
| $f_e$ | Elevating factor |
| $\tau$ | Timescale, s |
| **Superscripts** | |
| $^-$ | Reynolds average or filtered average |
| $^\sim$ | Favre average or Favre filtered average |
| **Subscripts** | |
| $t$ | Turbulent |
| $m$ | Specie |
| $c$ | Cut-off |
| $i$ | Integral |
| $k$ | Kolmogorov |
| $s$ | Smagorinsky |

## Appendix A. Chemical Timescale

The calculation method of the timescale for a complex chemical system is as follows: Firstly, the molar concentration of each specie is calculated by:

$$c_m = \frac{\rho Y_m}{W_m}, m = 1, \cdots, N \tag{A1}$$

where $W_m$ is the molar weight of specie $m$.

Secondly, for each reaction, the possible reverse rate is represented as:

$$\sum_{n=1}^{r_j} \gamma_n^j C_n^j \leftrightarrows \sum_{m=1}^{f_j} \gamma_m^{'j} C_m^{'j}, j = 1, \cdots, R \tag{A2}$$

where $R$ is the total number of reactions, $r_j$ is the number of reactants in reaction $j$, $\gamma_n^j$ is the reaction coefficient of reactant $n$ in reaction $j$, $f_j$ is the number of products of reaction $j$, and $\gamma_m^{'j}$ is the reaction coefficient of product $m$ in reaction $j$. Then, the system rate scale is estimated as the average of the reactions' rate scales weighted by their molar production rates, calculated by $W_{ave} = \frac{WR}{W}$, where $W$ and $WR$ are the summations of:

$$W = \sum_{j=1}^{R} \left( \sum_{m=1}^{f_j} \omega_f^j \gamma_m^j + \sum_{n=1}^{r_j} \omega_r^j \gamma_n^{'j} \right) \tag{A3}$$

and

$$WR = \sum_{j=1}^{R} \left[ \left( \sum_{m=1}^{f_j} \omega_f^j \gamma_m^j \right)^2 + \left( \sum_{n=1}^{r_j} \omega_r^j \gamma_n^{'j} \right)^2 \right] \tag{A4}$$

where $\omega_f^j$ and $\omega_r^j$ are the forward and reverse molar rates of reaction $j$, respectively.

Finally, the timescale of the chemical system is the ratio of the total molar concentration to the averaging system rate scale:

$$\tau_c = \frac{\sum_{m=1}^{N} c_m}{W_{ave}} \tag{A5}$$

This ensures that the dominant reaction provides the largest contribution to the system rate scale, and each reversible reaction is treated as two irreversible reactions including the forward and reverse reactions independently, so that there is no negative rate scale and timescale.

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
