# Peer review of "Investigation of Very Large Eddy Simulation Method for Applications of Supersonic Turbulent Combustion"

_aerospace, doi:10.3390/aerospace10040384_

Round 1

Reviewer 1 Report (Previous Reviewer 1)

I have gone through the revised manuscript version, and I have to say it was incredibly difficult to track the changes made to the manuscript by the authors since they are not highlighted in the revised manuscript as required.

I cannot recommend the publication since the authors have not adequately addressed my concerns regarding the grid independency study. Firstly, the grid independency or sensitivity study needs to be included in any numerical study so that the reader may be certain about the computations' validity. Then as the fundamental nature of the models used changes, so can the grid independency. 

For RANS, one typically shows how a characteristic variable of the problem at hand changes as the grid is progressively refined. In LES, if that is the case, it is also usual to compare the spectrum with a -5/3 slope, following Kolmogorov's universal equilibrium theory.

Fundamentally, the question is not if the paper's focus is grid independence/ sensitivity. These are fundamental test that any numerical work needs to adhere to in order to check if the obtained results are valid.

Author Response

Dear Reviewer:

Thank you for your review and comments on our manuscript (ID: aerospace-2306214). It will be beneficial for improving our manuscript and future studies.

We have studied the comments carefully and revised the manuscript according to your comments, including specific questions (SQ) and additional suggestions (AS). The revised portions are highlighted in the paper. The responses to each comment are shown as follows. We hope that this revision meets with approval.

Thank you very much!

Best wishes,

Sincerely,

Authors.

Reviewer 2 Report (Previous Reviewer 2)

In the resubmitted version, the authors have addressed nearly all my previous comments, except for those repeated below. I insist that they follow all comments properly rather than hush them up.

(1) The X-axis legend in Fig. 5 must be U/Uinf; the Y-legend in Fig. 6 must be P/Pinf.

(2) In experiments, the free stream turbulence was 1-2%. Is it taken into account in the authors calculations? I see the TKE and NUT scales in Figs. 8 to 10 start from 0.

(3) Is there any effect of water in vitiated air on the flow structure behind the strut?

(4) In the PaSR model, what is the effect of water in vitiated air on the chemical time scale tau_c and therefore on the reaction rate omega? This is worth to know as for the temperature level higher than 2200 K (see Figs. 18, 19, 21, 22) water dissociation could be important. In view of it, in addition to H and HO2, I would like to see the plots of water concentration in Fig. 20.

Author Response

Dear Reviewer:

Thank you for your review and comments on our manuscript (ID: aerospace-2306214). It will be beneficial for improving our manuscript and future studies.

We have studied the comments carefully and revised the manuscript according to your comments, including specific questions (SQ) and additional suggestions (AS). The revised portions are highlighted in the paper. The responses to each comment are shown as follows. We hope that this revision meets with approval.

Thank you very much!

Best wishes,

Sincerely,

Authors.

Reviewer 3 Report (Previous Reviewer 3)

The needful modifications are executed. Still few clarifications needed to be done. Please do the needful on the below suggestions.

1. The values mentioned in lines from 353 to 358 are good but references are needed. For the suitable CFL, the number is 200 for most of the cases but you have mentioned different values. Thus, please incorporate the additional content about your clarifications or add relevant references. 

2. If need, please incorporate relevant references at line 426. 

3. Please check and correct the Figure number on line 468. 

Author Response

Dear Reviewer:

Thank you for your review and comments on our manuscript (ID: aerospace-2306214). It will be beneficial for improving our manuscript and future studies.

We have studied the comments carefully and revised the manuscript according to your comments, including specific questions (SQ) and additional suggestions (AS). The revised portions are highlighted in the paper. The responses to each comment are shown as follows. We hope that this revision meets with approval.

Thank you very much!

Best wishes,

Sincerely,

Authors.

Round 2

Reviewer 1 Report (Previous Reviewer 1)

Thank you for your explanations. 

Validation is a fundamental aspect of CFD that always needs to be explained and shown to the reader so that there are no doubts surrounding the validity of the computations. Given that the authors addressed these aspects in the RANS and VLES components, I can recommend the manuscript for publication.

Reviewer 2 Report (Previous Reviewer 2)

The authors have addressed all my comments. The paper could be now considered for publication in Aerospace.

This manuscript is a resubmission of an earlier submission. The following is a list of the peer review reports and author responses from that submission.

Round 1

Reviewer 2 Report

This manuscript is focused on the performance of several RANS-based hybrid methods when applied to supersonic reactive flows. It fits the scope of Aerospace and could be considered for publication after major revision. My detailed comments are given below.

(1) Line 42: correction is needed to “the hybrid RNAS/LES.”

(2) Figure 3 needs the explicit scale along the Y-axis.

(3) Settles et al. indicated the measurement errors in their paper [56] attaining up to 10% in some flow regions. The authors must show the errors on their graphs. This could be instructive for estimating the strength of the various models.

(4) Figures 4 and 5 contain undefined parameters (Uinf and Pinf). The X-axis legends of these figures are not clear.

(5) Settles et al. [56] specify stagnation pressure and temperature rather than their static counterparts. The authors must refer to the original values with their uncertainties reported in [56].

(6) In Figure 6, the authors must define the Q-criterion.

(7) Caption of Fig. 7 does not contain (d).

(8) In [56], the free stream turbulence was 1-2%. Is it taken into account in the authors calculations? I see the TKE and NUT scales in Figs. 7 and 8 start from 0.

(9) Despite many speculations on the grid size control, the manuscript does not contain grid sensitivity tests. Those must be included.

(10) It would be instructive to have plots for the turbulent-to-laminar viscosity ratios for both test cases under study.

(11) Is there any effect of water in vitiated air on the flow structure behind the strut?

(12) In the PaSR model, what is the effect of water in vitiated air on the chemical time scale tau_c and therefore on the reaction rate omega? This is worth to know as for the temperature level higher than 2200 K (see Figs. 14, 15, 17) water dissociation could be important. In view of it, in addition to H and HO2, I would like to see the plots of water concentration in Fig. 16.

(13) The conclusions must be reformulated in a more definite way.

Reviewer 3 Report

First of all, the proposed work is good. Please consider the below suggestions for the further enhancement of the quality of this article. 

1. Since more abbreviations are imposed in this work, please form a separate table or section for "List of abbreviations" to create more clarity instead of confusion. 

2. Please incorporate relevant citations wherever governing equations are imposed in the main text. Also, suggest incorporating a separate table or section for "List of nomenclatures."

3. In line 259, the authors mentioned the term "inviscid fluxes" does the compressible turbulent flow contains an inviscid effect? Please, explain. 

4. What kind of computational procedure-based approach is imposed in this work? Implicit or Explicit? Please, explain your explanation for this selection. 

5. The boundary conditions mentioned in the lines from 313 to 316 are attained from where? is it extracted from reference 56?

6. The mesh explanations are good. But please list the mesh qualities based data such as aspect ratio, skewness, etc. 

7. Under boundary conditions, all the conditional data are relevant. But please explain the incorporation of periodic conditions in the adopted for the spanwise interfaces. 

8. In line 348, How this corresponding time step is achieved? This is a very low and complicated value. Generally, 1*10^-4 to 1*10^-5 based time steps are suitable for density-based solvers. Please explain your view about this doubt. 

9. What is the total number of time steps has been given for this computation?

10. Please name the solver tool imposed for this investigation.

11. A few typo errors are in the manuscript, so please check them. (line505)